# ST-TGExplainer: Disentangling Stability and Transition Patterns for Temporal GNN Interpretability

**Hongjiang Chen** [1]   **Xin Zheng** [2]   **Pengfei Jiao** [1]   **Huan Liu** [1]   **Zhidong Zhao** [1]
**Huaming Wu** [3]   **Feng Xia** [2]   **Shirui Pan** [4]

## Abstract

Temporal graph neural networks (TGNNs) have gained significant traction for solving real-world temporal graph tasks. However, their interpretability remains limited, as most TGNNs fail to identify which historical interactions most influence a given prediction. Despite promising progress on interpretable TGNNs, existing methods predominantly focus on previously seen historical interactions, which we term **stability** patterns, while overlooking newly emerging first-time interactions, which we term **transition** patterns. Both types of patterns are essential for faithful temporal explanations. To address this limitation, we propose **ST-TGEXPLAINER**, a self-explainable TGNN that disentangles **S**tability and **T**ransition patterns in temporal graphs for a more faithful **T**emporal **G**NN **E**XPLAINER. Guided by a disentangled information bottleneck objective, ST-TGEXPLAINER learns a compact explanatory subgraph that remains predictive of the event label while explicitly suppressing label-conditioned redundancy between stability and transition patterns. Extensive experiments demonstrate that ST-TGEXPLAINER achieves strong predictive performance and yields more faithful explanations. Code is available at https://github.com/hjchen-hdu/ST-TGExplainer.

## 1. Introduction

Temporal graph neural networks (TGNNs) have garnered considerable attention due to their applicability in real-world settings such as molecular biology (Ren et al., 2023), e-commerce platforms (Liu et al., 2025a), and social networks (Guo et al., 2025). These models are designed to capture both the topological structure and the dynamic dependencies that evolve over time within graph-structured data (Zhang et al., 2024). Despite their empirical success, TGNNs remain challenging to interpret: it is often unclear which historical interactions constitute the critical evidence for a predicted future event (Jiao et al., 2025). These challenges highlight the necessity of enhancing the explainability of TGNNs to bolster model trustworthiness, particularly in high-stakes fields such as healthcare, finance, and security (Amann et al., 2020; Zhang et al., 2026).

The goal of model explainability is to identify the patterns that drive a model's predictions (Schnake et al., 2021; Li et al., 2025a). Although this has been extensively studied for static GNNs (Yuan et al., 2021; Seo et al., 2023; Zhu et al., 2025), it does not readily transfer to temporal GNNs. In temporal graphs, interactions may occur at the same timestamp and in overlapping graph regions, complicating the dependency structure (Qin & Yeung, 2023; Chen et al., 2026b). Consequently, TGNN explanations should focus on historical interactions that are temporally proximate and spatially adjacent to the target event (Chen & Ying, 2023). In particular, an explanation should identify a compact set of historical interactions that drive a specific event prediction.

Recent efforts to explain TGNNs can be broadly classified into two lines: (a) post-hoc methods, such as T-GNNExplainer (Xia et al., 2023), TempME (Chen & Ying, 2023), and CoDy (Qu et al., 2025), which generate explanations after model training (Xia et al., 2025); and (b) self-explainable methods, such as TGIB (Seo et al., 2024), which integrate explanation mechanisms directly into the TGNN training process. Post-hoc methods are often computationally expensive and sensitive to perturbations in the TGNN architecture or the input temporal graph (Zhang et al., 2022). While self-explainable methods alleviate the efficiency issue, existing approaches still face two principal challenges: **Challenge-1 (Stability bias)**: Bias toward recurrent, previously seen interactions can dominate evidence attribution and obscure informative but rare transition signals; **Challenge-2 (Transition under-modeling)**: Inability

[1]Hangzhou Dianzi University, Hangzhou, China [2]RMIT University, Melbourne, Australia [3]Tianjin University, Tianjin, China [4]Griffith University, Gold Coast, Australia. Correspondence to: Pengfei Jiao <pjiao@hdu.edu.cn>.

*Proceedings of the 43rd International Conference on Machine Learning*, Seoul, South Korea. PMLR 306, 2026. Copyright 2026 by the author(s).

*Figure 1.* The MRR scores of seven popular TGNNs and our proposed ST-TGEXPLAINER ('Ours' in red bars) for predicting seen and unseen interactions on three datasets.

to effectively model newly emerging first-time interactions (i.e., unseen in history) yields unreliable explanations for novel events and behavioral shifts.

To further investigate these challenges, we conduct a comparative analysis of seven prevalent TGNNs. As illustrated in Figure 1, we evaluate their Mean Reciprocal Rank (MRR) across two categories of node interactions: repeated historical **seen** interactions (denoted by colored solid bars) vs. newly emerging **unseen** interactions (denoted by black hatched bars). The results reveal an essential observation: existing TGNNs generally perform well on repeated historical seen interactions, but struggle to capture newly emerging unseen ones. This gap suggests that faithful temporal explanations should account for both recurring (stability) evidence and newly emerging (transition) evidence, rather than being dominated by frequent interactions alone. Seen interactions capture recurring and stable patterns in temporal graphs, while new interactions often signify transitions, such as newly established relationships, community reorganization, or behavioral shifts between nodes (Grilli et al., 2017; Li et al., 2024; Zheng et al., 2026).

In light of this observation, we identify two patterns that play key roles in TGNN explanations, i.e., **stability** patterns for modeling seen historical interactions and **transition** patterns for characterizing newly emerging, previously unseen interactions indicative of dynamic behavioral shifts. Importantly, we propose **ST-TGEXPLAINER**, which disentangles **S**tability and **T**ransition patterns in temporal graphs for a more faithful **T**emporal **G**NN **EXPLAINER**. Guided by a disentangled information bottleneck objective, ST-TGEXPLAINER suppresses label-conditioned redundancy between stability and transition patterns while retaining predictive information. By jointly modeling historical dependencies and distinct behavioral patterns, our proposed ST-TGEXPLAINER supports fine-grained attribution while learning a compact explanatory subgraph that preserves label-relevant information. We conduct extensive experiments on link prediction, where ST-TGEXPLAINER outperforms existing methods on most datasets. Furthermore, evaluations using only the extracted explanatory subgraphs confirm that ST-TGEXPLAINER effectively captures label-

relevant information, ensuring faithful interpretability.

In summary, our main contributions are as follows:

- We propose ST-TGEXPLAINER, a self-explainable TGNN that disentangles stability and transition interaction evidence, enabling pattern-specific temporal explanations while maintaining predictive utility.

- We formulate a disentangled information bottleneck objective for temporal graph interpretability and derive tractable variational surrogates for its terms.

- We evaluate ST-TGEXPLAINER on multiple temporal graph benchmarks and show strong link prediction performance together with improved faithfulness of temporal explanations.

## 2. Related Work

**Temporal Graph Neural Networks.** Temporal graph neural networks (TGNNs) aim to model evolving node interactions over time, with future link prediction as a central task (Jiao et al., 2021; Chen et al., 2023; 2026a). Early models such as JODIE (Kumar et al., 2019), DyRep (Trivedi et al., 2019), and TGN (Rossi et al., 2020) adopt memory-based recurrent updates, whereas later approaches like TGAT (Xu et al., 2020), DyGFormer (Yu et al., 2023), and SALoM (Liu et al., 2025b) remove explicit memory modules and rely on attention or Transformer-based aggregation. GraphMixer (Cong et al., 2023), FreeDyG (Tian et al., 2024), and RepeatMixer (Zou et al., 2024) further leverage an efficient MLP-Mixer architecture to capture temporal dependencies with minimal complexity. Despite strong predictive performance, most existing TGNNs implicitly treat historical evidence as a homogeneous source when encoding temporal signals, which can cause the learned representations to be dominated by frequent, recurrent interactions (Poursafaei et al., 2022). Recent benchmarks such as TGB (Huang et al., 2023) and TGB-Seq (Yi et al., 2025) reveal a consistent generalization gap between recurrent (seen) and newly emerging (unseen) interactions, highlighting the need for designs that better account for distinct temporal interaction patterns.

**Explainability of GNNs.** Although GNNs have demonstrated effectiveness on graph-structured data, interpreting their predictions remains challenging due to complex architectures. A large body of work has investigated explainability for static graphs (Ying et al., 2019; Yuan et al., 2021; Zhu et al., 2025), but these methods cannot be directly applied to temporal graphs because the underlying evidence is time-dependent and dynamically evolving (Liu et al., 2023). Recently, several methods have been proposed to explain TGNNs, which can be broadly categorized into: (a) post-hoc explainers, e.g., T-GNNExplainer (Xia et al., 2023), TempME (Chen & Ying, 2023), GRExplainer (Li et al., 2025b), and CoDy (Qu et al., 2025), which generate explanations after model training and may suffer from inefficiency and sensitivity to changes in the base TGNN or temporal graph; and (b) self-explainable approaches, e.g., TGIB (Seo et al., 2024), which integrate explanation into training, typically by learning compact bottleneck subgraphs. While these methods can identify influential historical interactions, they usually treat temporal explanations uniformly and do not explicitly disentangle recurrent, seen interactions from rare or newly emerging, unseen interactions. Consequently, explanations can be biased toward stability explanations, which limits faithfulness in capturing emerging temporal dynamics. In this work, we mainly focus on self-explainable methods and address their key challenges: structural bias toward recurrent, seen historical interactions and inadequate modeling of newly emerging, unseen interactions.

## 3. Preliminaries and Problem Formulation

**Notations.** A temporal graph is represented as a sequence of non-decreasing chronological interactions denoted by $\mathcal{G} = \{(u_1, v_1, t_1), (u_2, v_2, t_2), \ldots, (u_k, v_k, t_k)\}$, where $0 \leq t_1 \leq t_2 \leq \cdots \leq t_k$. In this representation, $u_i$ and $v_i$ signify the source and destination nodes, respectively, for the $i$-th interaction occurring at timestamp $t_i$. The node set is denoted by $\mathcal{V}$, with each node $u \in \mathcal{V}$ associated with a node feature $\mathbf{x}_u \in \mathbb{R}^{d_N}$. The event set is denoted by $\mathcal{E} = \{e_i\}_{i=1}^K$, where $e_i = (u_i, v_i, t_i)$. Omitting subscript $i$, each interaction $e = (u, v, t)$ is characterized by an edge feature $\mathbf{x}_{u,v}^t \in \mathbb{R}^{d_E}$. Here, $d_N$ and $d_E$ represent the dimensions of node and edge features, respectively. The temporal graph $\mathcal{G}$ can be represented as a sequence of graphs at different timestamps, where each graph $\mathcal{G}^t$ captures the interactions that occur before timestamp $t$. Given $\mathcal{G}^t$ for the link prediction task, TGNNs aim to predict whether an interaction $e = (u, v, t)$ will occur, with a binary label $Y_{u,v}^t \in \{0, 1\}$, where $Y_{u,v}^t = 1$ indicates that $(u, v, t) \in \mathcal{E}$ and $Y_{u,v}^t = 0$ otherwise. For simplicity, we omit the subscripts $(u, v, t)$ of label $Y$ in the following discussion.

**Information Bottleneck.** Given a graph $\mathcal{G}^t$ and its associated label $Y$, the information bottleneck (IB) principle (Wu

et al., 2020) seeks to extract a compact subgraph $\mathcal{G}_E$ that preserves the information relevant to $Y$ while minimizing the dependency on the original graph $\mathcal{G}^t$ (Huang et al., 2026). This is formulated as the following optimization objective:

$$\min \ -I(Y; \mathcal{G}_E) + \beta I(\mathcal{G}^t; \mathcal{G}_E), \tag{1}$$

where $I(\cdot; \cdot)$ denotes mutual information, and $\beta$ is a Lagrange multiplier that balances prediction fidelity and compression. The first term preserves prediction fidelity by maximizing the mutual information between the graph label and the compressed subgraph, while the second term ensures compactness by minimizing the mutual information between the input graph and the compressed subgraph.

**Explanation for TGNNs.** Let $f$ be a well-trained TGNN that predicts whether an interaction $e = (u, v, t)$ occurs in $\mathcal{G}$. An explainer aims to extract a subset $\mathcal{G}_E \subseteq \mathcal{G}^t$ of influential historical interactions that drive the prediction made by $f$; this subset is referred to as the explanation.

The goal of explanation is to ensure that the selected subgraph $\mathcal{G}_E$ retains maximal information relevant to the model's prediction. In other words, the explanation should be both sufficient for reproducing the model's output and minimal in size, capturing only the most critical interactions.

## 4. Methodology

ST-TGEXPLAINER disentangles stability and transition patterns in temporal graphs to improve both link prediction performance and interpretability. As shown in Figure 2, ST-TGEXPLAINER comprises three key components: (1) **informative interaction selector**, which identifies influential historical interactions for the target event and produces an initial explanatory subgraph; (2) **stability–transition disentangler**, which disentangles the explanatory subgraph into complementary stability and transition patterns; (3) **explainable pattern feature ensembler**, which fuses the fine-grained interaction pattern representations for final prediction. Appendix B summarizes the training procedure and complexity analysis.

### 4.1. Disentangled IB on Temporal Graphs

Starting from the IB principle in Eq. (1), our goal is to extract an explanatory subgraph $\mathcal{G}_E$ in the input temporal graph $\mathcal{G}^t$ and predict a future interaction of $e = (u, v, t)$ with label $Y$. In contrast to existing approaches that rely on a single shared explanatory subgraph $\mathcal{G}_E$, which often leads to ambiguous and mixed predictive patterns that weaken interpretability, the proposed ST-TGEXPLAINER aims to further decompose the explanatory subgraph $\mathcal{G}_E = \{\mathcal{G}_S, \mathcal{G}_T\}$ into two distinct parts: a stability pattern $\mathcal{G}_S$ and a transition pattern $\mathcal{G}_T$. Under this disentangled perspective, we redevelop the IB objective on temporal GNN explainability

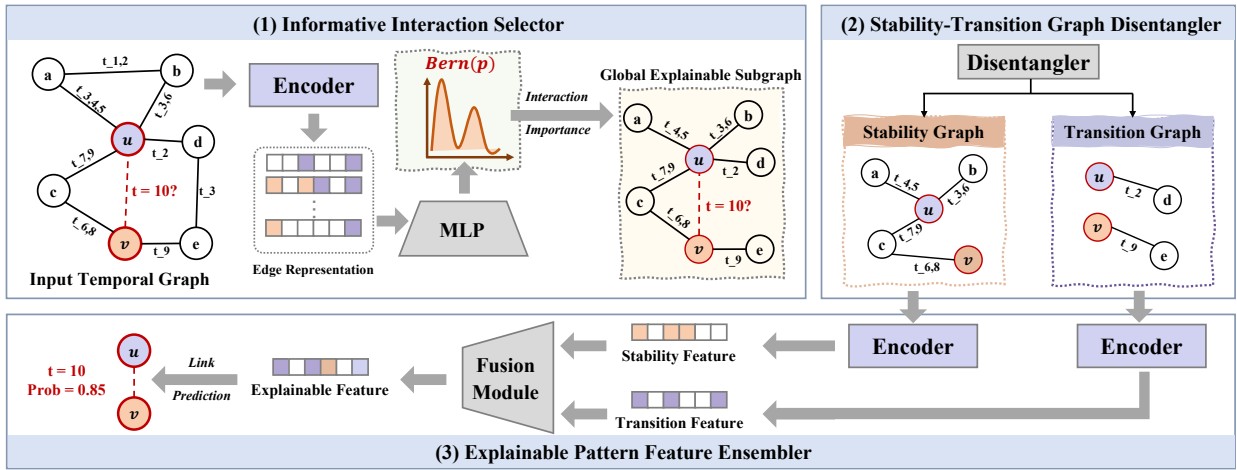

*Figure 2.* The overall framework of our method.

as:

$$\min -I(Y_S;\mathcal{G}_S) - I(Y_T;\mathcal{G}_T) + \beta\, I(\mathcal{G}^t;\mathcal{G}_S,\mathcal{G}_T), \quad (2)$$

where $Y_S$ and $Y_T$ are auxiliary variables denoting the hypothetical labels associated with the stability and transition patterns, respectively.

As we can only observe the event label $Y$ in real-world practice, both $\mathcal{G}_S$ and $\mathcal{G}_T$ tend to preserve information related to $Y$. This can cause them to encode redundant and overlapping evidence associated with the same label. To encourage the complementarity of explanations, we introduce a disentanglement regularization term $I(\mathcal{G}_S;\mathcal{G}_T|Y)$, which implicitly couples $Y$ with both $\mathcal{G}_S$ and $\mathcal{G}_T$ to promote their separation. Successful disentanglement implies that the encodings of $\mathcal{G}_S$ and $\mathcal{G}_T$ are conditionally independent given $Y$, thus capturing distinct and complementary semantics. Accordingly, we obtain the final disentangled IB objective as follows:

$$\min \underbrace{-I(Y;\mathcal{G}_S,\mathcal{G}_T)}_{\text{prediction}} + \beta\underbrace{I(\mathcal{G}^t;\mathcal{G}_S,\mathcal{G}_T)}_{\text{compression}} + \gamma\underbrace{I(\mathcal{G}_S;\mathcal{G}_T|Y)}_{\text{disentanglement}},$$

$$(3)$$

which consists of prediction, compression, and disentanglement components. Here, $\beta$ and $\gamma$ are hyperparameters. Since directly optimizing multivariate mutual information (MI) is difficult, we first analyze upper and lower bounds of all three MI terms to simplify the original objective, and then propose tractable variational upper bounds for each component. All theoretical details and proofs are provided in Appendix A.

**Proposition 4.1** (Variational upper bound of $-I(Y;\mathcal{G}_S,\mathcal{G}_T)$)**.** *Given the label $Y$, the stability–transition patterns $(\mathcal{G}_S,\mathcal{G}_T)$, and the fusion function $g_\phi$, let $U := (\mathcal{G}_S,\mathcal{G}_T)$. We have:*

$$-I(Y;U) = -\mathbb{E}_{p(Y,U)}\left[\log p(Y|U)\right] - H(Y),$$
$$\leq -\mathbb{E}_{p(Y,U)}\left[\log q_\theta(Y|g_\phi(U))\right] - H(Y), \quad (4)$$

*where $q_\theta(Y|g_\phi(U))$ is an approximation to the true posterior $p(Y|g_\phi(U))$. Since $H(Y)$ is constant, it is omitted from optimization.*

Accordingly, minimizing the prediction loss $\mathcal{L}_{\mathrm{Pre}} := -\mathbb{E}_{p(Y,U)}\left[\log q_\theta(Y|g_\phi(U))\right]$ serves as a variational surrogate for minimizing $-I(Y;\mathcal{G}_S,\mathcal{G}_T)$.

**Proposition 4.2** (Variational upper bound of $I(\mathcal{G}^t;\mathcal{G}_S,\mathcal{G}_T)$)**.** *Given the input temporal graph $\mathcal{G}^t$ and explanatory bottleneck $\mathcal{G}_E$, assume the Markov chain $\mathcal{G}^t \rightarrow \mathcal{G}_E \rightarrow (\mathcal{G}_S,\mathcal{G}_T)$, where $(\mathcal{G}_S,\mathcal{G}_T)$ are mappings of $\mathcal{G}_E$. Then:*

$$I(\mathcal{G}^t;\mathcal{G}_S,\mathcal{G}_T) \leq I(\mathcal{G}^t;\mathcal{G}_E)$$
$$\leq \mathbb{E}_{p(\mathcal{G}^t,\mathcal{G}_E)}\left[D_{\mathrm{KL}}\big(p_\theta(\mathcal{G}_E|\mathcal{G}^t)\,\|\,q(\mathcal{G}_E)\big)\right]$$
$$:= \mathcal{L}_{\mathrm{Com}},$$
$$(5)$$

*where $q(\mathcal{G}_E)$ is a variational approximation to the intractable marginal $p(\mathcal{G}_E)$.*

**Proposition 4.3** (Reformulation of $I(\mathcal{G}_S;\mathcal{G}_T|Y)$)**.** *Given the label $Y$ and the stability and transition patterns $(\mathcal{G}_S,\mathcal{G}_T)$, we have:*

$$I(\mathcal{G}_S;\mathcal{G}_T|Y) = \mathbb{E}_{p(Y)}\Big[D_{\mathrm{KL}}\big(p(\mathcal{G}_S,\mathcal{G}_T|Y)$$
$$\|\, p(\mathcal{G}_S|Y)\,p(\mathcal{G}_T|Y)\big)\Big]. \quad (6)$$

However, directly optimizing the KL divergence is intractable, as it requires evaluating high-dimensional conditional densities that are implicit under our neural generators. We therefore adopt a Jensen–Shannon (JS) divergence-based density-ratio formulation (Sugiyama et al., 2012) with a discriminator $d_\psi$ to approximate the conditional mutual information $I(\mathcal{G}_S;\mathcal{G}_T|Y)$. Specifically, the discriminator is trained to distinguish samples from the conditional joint $p(Y)p_\theta(\mathcal{G}_S,\mathcal{G}_T|Y)$ and the corresponding conditional product $p(Y)p_\theta(\mathcal{G}_S|Y)p_\theta(\mathcal{G}_T|Y)$. Positive samples $(\mathcal{G}_S,\mathcal{G}_T,Y$

are drawn using paired subgraphs under the same label, while negative samples $(\mathcal{G}_S, \tilde{\mathcal{G}}_T, Y)$ are constructed by independently resampling $\tilde{\mathcal{G}}_T \sim p_\theta(\mathcal{G}_T|Y)$. This leads to the following adversarial disentanglement objective:

$$
\begin{aligned}
\mathcal{L}_{\text{Dis}}(\theta, \psi) :=& \mathbb{E}_{p(\mathcal{G}_S, \mathcal{G}_T, Y)} \big[ \log d_\psi(\mathcal{G}_S, \mathcal{G}_T, Y) \big] \\
& + \mathbb{E}_{p(\mathcal{G}_S, \tilde{\mathcal{G}}_T, Y)} \big[ \log \big( 1 - d_\psi(\mathcal{G}_S, \tilde{\mathcal{G}}_T, Y) \big) \big].
\end{aligned}
\tag{7}
$$

The discriminator parameters $\psi$ are optimized to maximize $\mathcal{L}_{\text{Dis}}$, while the encoder and disentangler parameters $\theta$ are optimized to minimize it. This adversarial optimization encourages the conditional joint to match the conditional product in the JS sense, thereby promoting conditional independence between $\mathcal{G}_S$ and $\mathcal{G}_T$ given $Y$.

**Final Objective.** Combining the variational surrogates derived above, the overall training objective of our model is formulated as a weighted sum of three components: a prediction term that preserves label-relevant information, a compression term that enforces a compact explanatory bottleneck, and a disentanglement term that suppresses conditional redundancy between the stability and transition patterns. Specifically, we solve the following optimization problem:

$$
\min_{\theta, \phi} \max_\psi \ \mathcal{L}_{\text{Pre}}(\theta, \phi) + \beta \, \mathcal{L}_{\text{Com}}(\theta) + \gamma \, \mathcal{L}_{\text{Dis}}(\theta, \psi), \tag{8}
$$

where $\beta$ and $\gamma$ control the trade-off between prediction fidelity, information compression, and stability–transition disentanglement.

## 4.2. Informative Interaction Selector

To mitigate the computational overhead of modeling the full temporal graph $\mathcal{G}^t$, we construct a compact ego-temporal subgraph around the queried event $e = (u, v, t)$. Specifically, we collect historical interactions incident to either $u$ or $v$ that occurred before $t$, and retain the $L$ most recent ones to form the model input $\mathcal{G}_{u,v}^t$. This sampled subgraph captures the most relevant recent evidence while keeping computation tractable; any subset of its historical interactions can serve as a candidate explanation.

### 4.2.1. ENCODER.

Given the input subgraph $\mathcal{G}_{u,v}^t$, we first extract node and edge features, denoted as $\mathbf{X}_N^t \in \mathbb{R}^{L \times d_N}$ and $\mathbf{X}_E^t \in \mathbb{R}^{L \times d_E}$, respectively. For temporal information, we follow (Xu et al., 2020) to map timestamps into temporal features $\mathbf{X}_T^t = [\phi(t - t_1), \dots, \phi(t - t_L)] \in \mathbb{R}^{L \times d_T}$, where the function $\phi(\Delta t) = [\cos(w_1 \Delta t), \dots, \cos(w_{d_T} \Delta t)]$ and $\{w_i\}_{i=1}^{d_T}$ is a set of learnable weights. Here, $\Delta t$ represents the time difference between the current timestamp $t$ and the timestamps of past events, capturing periodic temporal patterns. Additionally, we construct a relative time encoding $\mathbf{X}_R^t =$

$\varphi([\delta t(u), \delta t(v)]) \in \mathbb{R}^{L \times d_R}$, where $\delta t(u) = t - t^-(u)$ denotes the elapsed time since node $u$'s last activity at timestamp $t^-(u)$, and similarly for $\delta t(v)$. The function $\varphi(\cdot)$ is a trainable mapping that transforms relative times into a $d_R$-dimensional space, enabling the model to encode the recent activity patterns of both source and destination nodes. We then concatenate all feature components and project them through an MLP to obtain the unified feature sequence:

$$
\mathbf{Z}^{(0)} = \text{MLP}\left([\mathbf{X}_N^t, \mathbf{X}_E^t, \mathbf{X}_T^t, \mathbf{X}_R^t]\right) \in \mathbb{R}^{L \times d}. \tag{9}
$$

To model structural and temporal dependencies, we apply $l$ stacked layers of MLP-Mixer (Tolstikhin et al., 2021). Each layer updates the representation as follows:

$$
\begin{aligned}
\tilde{\mathbf{Z}}^{(l)} &= \mathbf{Z}^{(l-1)} + \mathbf{W}_1^{(l)} \text{GeLU}\left(\mathbf{W}_2^{(l)} \text{LN}\left(\mathbf{Z}^{(l-1)}\right)\right), \\
\mathbf{Z}^{(l)} &= \tilde{\mathbf{Z}}^{(l)} + \mathbf{W}_3^{(l)} \text{GeLU}\left(\mathbf{W}_4^{(l)} \text{LN}\left(\tilde{\mathbf{Z}}^{(l)}\right)\right),
\end{aligned}
\tag{10}
$$

where GeLU is the activation function, LN denotes layer normalization, and $\mathbf{W}_*^{(l)}$ are learnable parameters for the $l$-th layer. Finally, we define the encoder output edge representation as:

$$
\mathbf{H}^t = \text{Encoder}_\theta(\mathcal{G}_{u,v}^t) := \mathbf{Z}^{(l)} \in \mathbb{R}^{L \times d}. \tag{11}
$$

### 4.2.2. STOCHASTIC EXPLANATORY SUBGRAPH.

We introduce a parameterized bottleneck $p_\theta(\mathcal{G}_E|\mathcal{G}_{u,v}^t)$ to model the conditional distribution of the explanatory subgraph. To enable tractable optimization, we adopt a factorized multivariate Bernoulli distribution:

$$
p_\theta(\mathcal{G}_E|\mathcal{G}_{u,v}^t) = \prod_{e \in \mathcal{G}_E} p_e \prod_{e \in \mathcal{G}_{u,v}^t \setminus \mathcal{G}_E} (1 - p_e), \tag{12}
$$

where $p_e$ denotes the inclusion probability of edge $e$ in the explanatory subgraph $\mathcal{G}_E$. To enable gradient-based learning with discrete edge selections, we use the Gumbel-Softmax/Concrete relaxation (Jang et al., 2017) to obtain a differentiable approximation of Bernoulli sampling. Specifically, the probability $p_e$ is parameterized as:

$$
p_e = \sigma\left(\text{MLP}(\mathbf{h}_e)\right), \tag{13}
$$

where $\sigma(\cdot)$ is the sigmoid function, and $\mathbf{h}_e$ is the edge representation computed by the encoder $\text{Encoder}_\theta$ (i.e., the row of $\mathbf{H}^t$ corresponding to $e$). A higher $p_e$ indicates a greater likelihood that edge $e$ contributes to $\mathcal{G}_E$. This formulation treats the inclusion of each edge as an independent Bernoulli trial, thereby simplifying mutual information estimation.

For the variational marginal $q(\mathcal{G}_E)$ (i.e., a simple prior over explanations), we assume a multivariate Bernoulli distribution with a fixed inclusion rate $r \in [0, 1]$ shared across edges:

$$
q(\mathcal{G}_E) = r^{|\mathcal{G}_E|} (1 - r)^{|\mathcal{G}_{u,v}^t| - |\mathcal{G}_E|}. \tag{14}
$$

The resulting information loss, corresponding to the KL divergence between the two distributions, admits a closed-form decomposition:

$$\mathcal{L}_{\text{Com}} = \mathbb{E}_{p(\mathcal{G}_{u,v}^t)}\Big[ D_{\text{KL}}\big(p_\theta(\mathcal{G}_E|\mathcal{G}_{u,v}^t) \,\|\, q(\mathcal{G}_E)\big)\Big]$$

$$= \mathbb{E}_{p(\mathcal{G}_{u,v}^t)}\left[ \sum_{e \in \mathcal{G}_{u,v}^t} p_e \log \frac{p_e}{r} + (1 - p_e)\log\frac{1 - p_e}{1 - r}\right].$$
(15)

This objective penalizes excessive reliance on the input graph $\mathcal{G}_{u,v}^t$, encouraging the model to retain only the most informative edges in $\mathcal{G}_E$.

### 4.3. Stability–Transition Graph Disentangler

We disentangle the explanatory subgraph $\mathcal{G}_E$ by leveraging empirical interaction frequency as a soft proxy for recurrence. As illustrated in Figure 2, node $u$ interacts twice with node $a$, indicating stable evidence, whereas the interaction between $u$ and $d$ occurs only once, suggesting weaker recurrence and a stronger transition signal. Conceptually, transition patterns correspond to newly emerging interactions, but hard first-time indicators are brittle under neighbor sampling and do not provide a smooth optimization signal. We therefore use frequency to construct a differentiable soft assignment rather than redefining transition solely as low frequency. Let $\mathbf{h}_F \in \mathbb{R}^L$ denote the frequency vector over the sampled interaction history $\mathcal{G}_{u,v}^t$, where the $i$-th entry is the number of occurrences of the corresponding node pair. We then transform $\mathbf{h}_F$ into a soft assignment score $\mathbf{p}_f \in [0,1]^L$ via a neural network:

$$\mathbf{p}_f = \sigma(\text{MLP}(\mathbf{h}_F)),$$
(16)

Based on $\mathbf{p}_f$, we derive the stability and transition subgraphs by applying element-wise masking to the explanatory subgraph:

$$\mathcal{G}_S = \mathbf{p}_f \odot \mathcal{G}_E, \quad \mathcal{G}_T = (1 - \mathbf{p}_f) \odot \mathcal{G}_E,$$
(17)

where $\odot$ denotes the Hadamard product. By construction, this yields a soft additive decomposition of $\mathcal{G}_E$ at the edge-weight level: $\mathcal{G}_S$ gives larger weight to recurrent interactions, while $\mathcal{G}_T$ emphasizes edges with weaker historical support, including first-time or rare interactions. To encode these subgraphs, we employ a shared encoder and aggregate node-level features via mean pooling:

$$\mathbf{h}_S = \text{Mean}(\text{Encoder}_\theta(\mathcal{G}_S)) \in \mathbb{R}^d,$$
$$\mathbf{h}_T = \text{Mean}(\text{Encoder}_\theta(\mathcal{G}_T)) \in \mathbb{R}^d.$$
(18)

To encourage a complementary decomposition between stability and transition patterns, we introduce conditional JS-divergence-based adversarial regularization on the learned representations. Specifically, we construct negative pairs by conditionally resampling one component, i.e.,

*Table 1.* Statistics of datasets used for experiments.

| Dataset | #Nodes | #Edges | Repeat ratio | Duration |
|---|---|---|---|---|
| Wikipedia | 9,227 | 157,474 | 88.41% | 1 month |
| Reddit | 10,984 | 672,447 | 88.32% | 1 month |
| UCI | 1,899 | 58,835 | 66.06% | 196 days |
| Enron | 184 | 125,235 | 90.79% | 3 years |
| USLegis | 225 | 60,396 | 56.25% | 12 congresses |
| Can.Parl. | 734 | 74,478 | 31.08% | 14 years |

$(\mathbf{h}_S, \tilde{\mathbf{h}}_T, Y) \sim p_\theta(\mathbf{h}_S|Y)\,p_\theta(\mathbf{h}_T|Y)$, where $\tilde{\mathbf{h}}_T$ denotes an independent draw under the same label $Y$. We define the disentanglement objective as the binary cross-entropy loss of the discriminator $d_\psi$:

$$\mathcal{L}_{\text{Dis}}(\theta,\psi) = -\mathbb{E}_{p(\mathcal{G}_S,\mathcal{G}_T,Y)} \log d_\psi(\mathbf{h}_S, \mathbf{h}_T, Y)$$
$$-\mathbb{E}_{p(\mathcal{G}_S,\tilde{\mathcal{G}}_T,Y)} \log\big(1 - d_\psi(\mathbf{h}_S, \tilde{\mathbf{h}}_T, Y)\big),$$
(19)

which enforces conditional independence between $\mathbf{h}_S$ and $\mathbf{h}_T$ given $Y$ through adversarial training. This representation-level objective provides a tractable surrogate for the subgraph-level disentanglement term in Proposition 4.3, and directly suppresses label-conditioned redundancy in the predictive representations used by the TGNN.

### 4.4. Explainable Pattern Feature Ensembler

To predict the interaction label $\hat{Y}$ from disentangled temporal patterns, we design a pattern feature ensembler that integrates stability and transition representations. Let $g_\phi$ denote a fusion function over the latent vectors $\mathbf{h}_S$ and $\mathbf{h}_T$:

$$\mathbf{h}_E = \text{MLP}(\mathbf{h}_S \| \mathbf{h}_T) \in \mathbb{R}^d,$$
(20)

where $\|$ indicates vector concatenation, and $\mathbf{h}_E$ denotes the fused embedding capturing explainable graph-level semantics. Subsequently, the interaction probability $\hat{Y}$ is computed using a prediction head composed of a two-layer MLP with a ReLU activation followed by a sigmoid function:

$$\hat{Y} = \sigma\left(\text{MLP}(\text{ReLU}(\text{MLP}(\mathbf{h}_E)))\right),$$
(21)

where $\sigma(\cdot)$ is the sigmoid function. To optimize the model for link prediction, we employ the binary cross-entropy loss:

$$\mathcal{L}_{\text{Pre}} = -\mathbb{E}_{p(Y,\mathcal{G}_{u,v}^t)}\Big[ Y \log \hat{Y} + (1 - Y)\log(1 - \hat{Y})\Big],$$
(22)

where $Y$ denotes the ground-truth label indicating whether $(u, v, t)$ exists in $\mathcal{G}$, and $\hat{Y}$ denotes the predicted value.

## 5. Experiments

### 5.1. Experimental Setup

**Datasets.** We evaluate ST-TGEXPLAINER on six real-world temporal graph datasets spanning diverse application domains (e.g., social and political networks). Dataset

*Table 2.* Link prediction performance AP (%) comparison of our proposed ST-TGEXPLAINER (Ours) with nine baseline methods.

| Models | Wikipedia | Reddit | UCI | Enron | USLegis | Can.Parl. |
|---|---|---|---|---|---|---|
| JODIE | 94.62 ± 0.50 | 97.11 ± 0.30 | 86.73 ± 1.00 | 77.31 ± 4.20 | 73.31 ± 0.40 | 69.26 ± 0.31 |
| DyRep | 92.43 ± 0.37 | 96.09 ± 0.11 | 53.67 ± 2.10 | 74.55 ± 3.95 | 57.28 ± 0.71 | 54.02 ± 0.76 |
| TGAT | 95.34 ± 0.10 | 98.12 ± 0.20 | 73.01 ± 0.60 | 68.02 ± 0.10 | 68.89 ± 1.30 | 70.73 ± 0.72 |
| TGN | 97.58 ± 0.20 | 98.30 ± 0.20 | 80.40 ± 1.40 | 79.91 ± 1.30 | 75.13 ± 1.30 | 70.88 ± 2.34 |
| CAWN | 98.28 ± 0.20 | 97.95 ± 0.20 | 90.03 ± 0.40 | 89.56 ± 0.09 | 69.94 ± 0.40 | 69.82 ± 2.34 |
| GraphMixer | 97.25 ± 0.03 | 97.31 ± 0.01 | 93.25 ± 0.57 | 82.25 ± 0.16 | 70.74 ± 1.02 | 77.04 ± 0.46 |
| DyGFormer | 99.03 ± 0.02 | 99.22 ± 0.01 | 95.79 ± 0.17 | 92.47 ± 0.12 | 71.11 ± 0.59 | 76.30 ± 2.56 |
| FreeDyG | 99.20 ± 0.02 | 99.40 ± 0.01 | 96.28 ± 0.11 | 90.29 ± 0.10 | 69.66 ± 0.63 | 72.22 ± 2.47 |
| TGIB | **99.37 ± 0.09** | - | 93.60 ± 0.24 | 82.42 ± 0.11 | **91.61 ± 0.34** | 87.07 ± 0.44 |
| **Ours** | 99.21 ± 0.05 | **99.43 ± 0.01** | **97.28 ± 0.51** | **94.87 ± 0.59** | **94.47 ± 0.89** | **90.25 ± 0.48** |

statistics are summarized in Table 1, and further details are provided in Appendix C.

**Baselines.** For link prediction, we compare against nine representative TGNNs: JODIE (Kumar et al., 2019), DyRep (Trivedi et al., 2019), TGAT (Xu et al., 2020), TGN (Rossi et al., 2020), CAWN (Wang et al., 2021), Graph-Mixer (Cong et al., 2023), DyGFormer (Yu et al., 2023), FreeDyG (Tian et al., 2024), and TGIB (Seo et al., 2024).

For explainability, we compare ACC-AUC against seven representative explanation methods: Random, Grad-CAM (Pope et al., 2019), GNNExplainer (Ying et al., 2019), PGExplainer (Luo et al., 2020), T-GNNExplainer (Xia et al., 2023), TempME (Chen & Ying, 2023), and TGIB (Seo et al., 2024). Random produces explanations by uniformly sampling nodes under the sparsity constraint. For post-hoc explainers, we adopt GraphMixer (Cong et al., 2023) as the backbone model. We further include CoDy (Qu et al., 2025) in the AUFSC-based fidelity evaluation in Appendix F.1. Additional baseline details are provided in Appendix D.

**Metrics and Implementation.** For link prediction, we evaluate performance using Average Precision (AP) and Mean Reciprocal Rank (MRR), which are widely adopted in the literature (Yi et al., 2025). For explainability, we follow (Chen & Ying, 2023) and report ACC-AUC, defined as the AUC of prediction accuracy as the explanation sparsity varies from 0 to 0.3 with a step size of 0.002. We additionally report AUFSC (Qu et al., 2025) in Appendix F.1 to evaluate necessity and sufficiency across sparsity levels. Missing entries ("-") correspond to out-of-memory failures on a 24GB GPU. Additional implementation details and hyperparameter settings are provided in Appendix E.

### 5.2. Link Prediction Performance

We evaluate link prediction performance using AP over all six datasets and MRR on datasets with more than 1,000 nodes (Wikipedia, Reddit, and UCI), where MRR is computed by ranking the true destination among 100 sampled negative destinations. Taken together, Tables 2 and 3 show

*Table 3.* Link prediction performance MRR (%) comparison of ST-TGEXPLAINER (Ours) with completed baseline methods.

| Models | Wikipedia | Reddit | UCI |
|---|---|---|---|
| JODIE | 76.48 ± 1.72 | 77.16 ± 1.27 | 51.43 ± 1.74 |
| DyRep | 67.42 ± 4.21 | 72.33 ± 2.14 | 14.00 ± 2.71 |
| TGAT | 72.72 ± 1.50 | 78.03 ± 0.25 | 31.50 ± 0.48 |
| TGN | 82.22 ± 0.39 | 79.25 ± 0.24 | 45.29 ± 6.43 |
| CAWN | 86.07 ± 0.08 | 87.66 ± 0.04 | 66.96 ± 0.48 |
| GraphMixer | 73.57 ± 1.48 | 70.87 ± 0.38 | 59.58 ± 1.00 |
| DyGFormer | 88.69 ± 0.04 | 88.70 ± 0.05 | 76.61 ± 0.26 |
| FreeDyG | **89.00 ± 0.94** | 87.60 ± 0.23 | 73.60 ± 0.78 |
| **Ours** | 86.68 ± 0.05 | **88.96 ± 0.45** | **81.68 ± 0.43** |

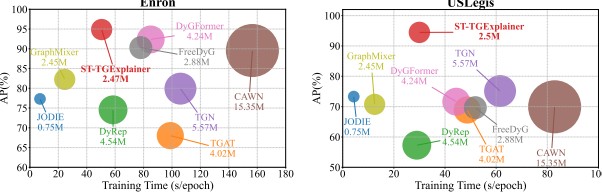

*Figure 3.* Comparison of model performance, parameter size, and training time per epoch on Enron and USLegis.

that ST-TGEXPLAINER improves link prediction from both classification and ranking perspectives. In terms of AP, ST-TGEXPLAINER achieves the best performance on five of the six datasets and remains competitive on Wikipedia, indicating that the learned explanatory subgraph preserves label-relevant information across diverse temporal graph regimes. The gains are especially pronounced on low-repeat datasets such as USLegis and Can.Parl., suggesting that the stability–transition disentanglement helps model interactions that cannot be explained by recurrence alone. The MRR results provide a complementary view: ST-TGEXPLAINER achieves the best ranking performance on Reddit and UCI and obtains a particularly large margin on UCI, where newly emerging interactions are more frequent and ranking the true destination among hard negatives is more challenging. On Wikipedia, ST-TGEXPLAINER remains close to the

*Table 4.* Explanation performance ACC-AUC (%) comparison of ST-TGEXPLAINER (Ours) with seven baseline methods.

| Models | Wikipedia | Reddit | UCI | Enron | USLegis | Can.Parl. |
|---|---|---|---|---|---|---|
| Random | 77.31 ± 2.37 | 85.08 ± 0.72 | 53.56 ± 1.27 | 64.07 ± 0.86 | 85.54 ± 0.93 | 87.79 ± 0.51 |
| Grad-CAM | 76.63 ± 0.01 | 84.44 ± 0.41 | 82.64 ± 0.01 | 72.50 ± 0.01 | 88.98 ± 0.01 | 85.80 ± 0.01 |
| GNNExplainer | 89.21 ± 0.63 | 95.10 ± 0.36 | 61.02 ± 0.37 | 74.23 ± 0.13 | 89.67 ± 0.35 | 92.28 ± 0.10 |
| PGExplainer | 85.19 ± 1.24 | 92.46 ± 0.42 | 63.76 ± 1.06 | 75.39 ± 0.43 | 92.37 ± 0.10 | 90.63 ± 0.32 |
| T-GNNExplainer | 87.69 ± 0.86 | **95.82 ± 0.73** | 80.47 ± 0.87 | 81.87 ± 0.45 | 93.04 ± 0.45 | 93.78 ± 0.74 |
| TempME | 90.15 ± 0.30 | 95.05 ± 0.19 | 87.06 ± 0.12 | 79.69 ± 0.33 | 95.00 ± 0.16 | 95.98 ± 0.21 |
| TGIB | 88.09 ± 0.68 | - | 87.06 ± 1.04 | 83.55 ± 0.91 | 93.33 ± 0.72 | 89.72 ± 1.18 |
| **Ours** | **92.47 ± 0.26** | 95.39 ± 0.11 | **96.04 ± 0.39** | **90.15 ± 0.44** | **97.49 ± 0.38** | **96.88 ± 0.87** |

strongest ranking baselines while maintaining strong AP and favorable seen/unseen behavior in Figure 1. These results support two conclusions: the disentanglement module improves predictive accuracy by capturing both recurrent and emerging temporal patterns, and the information bottleneck objective helps suppress label-irrelevant history while retaining salient evidence for future interactions. TGIB is excluded from the MRR comparison because it did not finish within 24 hours under the MRR protocol.

Figure 3 compares AP, training time per epoch (seconds), and parameter size (MB) on Enron and USLegis. Walk-based models such as CAWN are computationally intensive, whereas memory-based methods (e.g., DyRep and JODIE) are more efficient but underperform. In contrast, ST-TGEXPLAINER attains top accuracy with a compact parameter footprint and moderate training cost.

### 5.3. Explanation Performance

Table 4 reports the explanation performance. We make three observations. First, ST-TGEXPLAINER outperforms most baselines across all datasets, demonstrating its ability to identify explanation subgraphs that preserve the prediction behavior of the original model. Second, on datasets with low repeat ratios (e.g., UCI, USLegis, and Can.Parl.), ST-TGEXPLAINER achieves up to a 9% improvement over competing methods, which supports the conjecture that many existing explainers are biased toward repeated interactions and therefore under-emphasize exploratory evidence. Third, Grad-CAM underperforms the Random baseline on some benchmarks, suggesting limited robustness and unreliable explanations.

Beyond prediction consistency measured by ACC-AUC, we further evaluate necessity and sufficiency using AUFSC (Qu et al., 2025); the results in Appendix F.1 show that the selected edges provide both necessary and sufficient evidence for the model's predictions. Qualitative case studies in Appendix F.5 further show that ST-TGEXPLAINER separates stability and transition evidence more cleanly than post-hoc baselines. Overall, ST-TGEXPLAINER exhibits stable and theoretically grounded explanation performance.

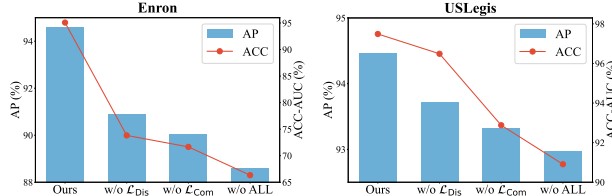

*Figure 4.* Ablation study results in terms of link prediction AP (%) and explanation ACC-AUC (%).

*Table 5.* Ablation results on Wikipedia and UCI.

| Variant | Wikipedia | | UCI | |
|---|---|---|---|---|
| | AP | ACC-AUC | AP | ACC-AUC |
| Full model | 99.21 | 92.47 | 97.28 | 96.04 |
| $\mathcal{G}_E$-only | 98.14 | 88.58 | 96.06 | 88.06 |
| $\mathcal{G}_S$-only | 98.01 | 87.93 | 91.76 | 84.67 |
| $\mathcal{G}_T$-only | 96.36 | 77.61 | 93.62 | 85.18 |

### 5.4. Ablation Study

We conduct ablation studies to evaluate the contribution of each component in ST-TGEXPLAINER. As shown in Figure 4, we consider three variants: "w/o $\mathcal{L}_{\mathrm{Dis}}$", which removes the discrimination loss for stability–transition disentanglement; "w/o $\mathcal{L}_{\mathrm{Com}}$", which removes the compression loss; and "w/o ALL", which retains only the encoder and explanation modules. We make three observations: (1) Removing any component degrades performance on both prediction and explanation tasks. (2) Eliminating $\mathcal{L}_{\mathrm{Dis}}$ markedly reduces prediction accuracy, whereas removing $\mathcal{L}_{\mathrm{Com}}$ substantially impairs explanation quality, suggesting that $\mathcal{L}_{\mathrm{Dis}}$ is critical for accuracy while $\mathcal{L}_{\mathrm{Com}}$ is essential for faithful explanations. (3) The ensemble module improves both link prediction and explanation performance by effectively integrating representations from stability and transition patterns. Additional ablations in Appendix F.3 extend these comparisons to four more datasets and show the same trend.

We further isolate the contribution of the proposed stability–transition decomposition in Table 5. The $\mathcal{G}_E$-only variant removes the split and predicts directly from the mixed expla-

nation subgraph $\mathcal{G}_E$, while the $\mathcal{G}_S$-only and $\mathcal{G}_T$-only variants retain only one branch. This comparison tests whether the gain comes from the decomposition itself rather than merely from an additional explanatory bottleneck. The branch-specific results align with dataset recurrence: $\mathcal{G}_S$-only performs better on Wikipedia, whose repeat ratio is 88.41%, whereas $\mathcal{G}_T$-only is relatively stronger on UCI, whose repeat ratio is 66.06%. The full model surpasses $\mathcal{G}_E$-only by +3.89 and +7.98 ACC-AUC points on Wikipedia and UCI, respectively, indicating that stability and transition patterns provide complementary evidence beyond a single mixed explanation subgraph.

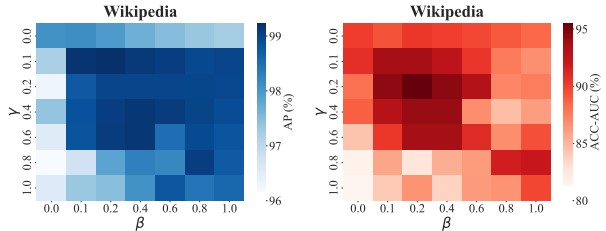

*Figure 5.* Hyperparameter sensitivity analysis of ST-TGEXPLAINER on the Wikipedia dataset in terms of link prediction AP (%) and explanation ACC-AUC (%).

## 5.5. Hyperparameter Sensitivity

We conduct a sensitivity analysis on Wikipedia to examine the effects of the hyperparameters $\beta$ and $\gamma$, which weight $\mathcal{L}_{\text{Com}}$ and $\mathcal{L}_{\text{Dis}}$, respectively, over the range $[0.0, 1.0]$. As shown in Figure 5, setting either hyperparameter to zero substantially degrades both link prediction (AP) and explainability (ACC-AUC), underscoring the necessity of the compression and disentanglement objectives. Without sufficient compression, the selected subgraph may retain redundant historical interactions; without disentanglement, stability and transition evidence can become less separated. Increasing $\beta$ and $\gamma$ initially improves performance, which peaks around $\beta = 0.1$ and $\gamma \in \{0.1, 0.2\}$, and then declines, likely due to over-regularization that removes useful predictive information. Overall, strong performance requires balancing mutual-information-based compression against the disentanglement loss.

## 5.6. Visualization

To assess the effectiveness of our disentanglement approach, Figure 6 shows PCA projections of the learned latent vectors $\mathbf{h}_S$, $\mathbf{h}_T$, and $\mathbf{h}_E$ on the Wikipedia and USLegis datasets. Here, $\mathbf{h}_S$ and $\mathbf{h}_T$ are produced by the disentanglement module, whereas $\mathbf{h}_E$ denotes the representation learned from the explanatory graph without the stability–transition split. The separation between $\mathbf{h}_S$ and $\mathbf{h}_T$ suggests that ST-TGEXPLAINER captures distinct temporal patterns, in contrast to the more entangled representation $\mathbf{h}_E$. This qual-

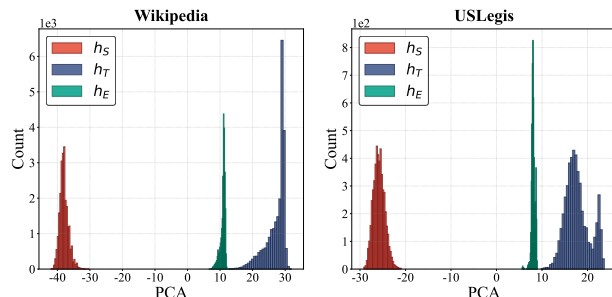

*Figure 6.* Embedding distributions of $\mathbf{h}_S$ (stability graph), $\mathbf{h}_T$ (transition graph), and $\mathbf{h}_E$ (explanatory graph).

itative evidence is consistent with the ablation results, where using both stability and transition branches yields better performance than relying on a single mixed representation. Additional results are reported in Appendix F.4.

## 6. Conclusion

We present ST-TGEXPLAINER, a self-explainable TGNN that leverages a disentangled information bottleneck to extract compact, predictive subgraphs while explicitly separating stability and transition evidence. Across multiple benchmarks, ST-TGEXPLAINER achieves strong link prediction accuracy and improved explanation faithfulness.

**Limitation.** Despite its strong empirical performance, several limitations need to be noted. First, it relies on one-hop histories and may miss higher-order temporal dependencies; extending to multi-hop neighborhoods may improve fidelity but increase computational cost and hinder scalability. Second, the frequency-based disentangler relies on heuristics (e.g., frequency estimation and soft assignment), and its robustness may vary across datasets and temporal regimes. We leave efficient multi-hop designs and more principled, scalable frequency estimation to future work.

## Impact Statement

This paper presents work whose goal is to advance the field of Machine Learning. There are many potential societal consequences of our work, none of which we feel must be specifically highlighted here.

## Acknowledgements

Pengfei Jiao was partially supported by the National Natural Science Foundation of China under Grant 62372146, the Zhejiang Province Key R&D Program under Grants 2025C01023 and 2024C01102, and the Zhejiang Provincial Key Laboratory for Sensitive Data Security Protection and Confidentiality Management under Grant 2024E10048. Huaming Wu was supported by the Tianjin Natural Science Foundation Project under Grant 25JCYBJC01540.

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

# A. Variational Bounds for the IB Objective

This appendix provides a detailed and fully expanded proof of the variational upper bounds used to derive the tractable optimization objective in Eq. (8). In particular, the derivations below expand and justify the bounds stated in Propositions 4.1–4.3 in the main text, while making explicit every probabilistic step.

## A.1. Prediction term: variational surrogate for $-I(Y; \mathcal{G}_S, \mathcal{G}_T)$.

We derive a variational upper bound for $-I(Y; \mathcal{G}_S, \mathcal{G}_T)$. Let us begin from the definition of mutual information between the label $Y$ and the stability–transition patterns $(\mathcal{G}_S, \mathcal{G}_T)$:

$$I(Y; \mathcal{G}_S, \mathcal{G}_T) := \mathbb{E}_{p(Y, \mathcal{G}_S, \mathcal{G}_T)} \left[ \log \frac{p(Y, \mathcal{G}_S, \mathcal{G}_T)}{p(Y)\, p(\mathcal{G}_S, \mathcal{G}_T)} \right]. \tag{23}$$

Using the chain rule of probability, we have $p(Y, \mathcal{G}_S, \mathcal{G}_T) = p(Y|\mathcal{G}_S, \mathcal{G}_T) p(\mathcal{G}_S, \mathcal{G}_T)$. Substituting this into the log-ratio, we obtain:

$$\log \frac{p(Y, \mathcal{G}_S, \mathcal{G}_T)}{p(Y)\, p(\mathcal{G}_S, \mathcal{G}_T)} = \log \frac{p(Y|\mathcal{G}_S, \mathcal{G}_T) p(\mathcal{G}_S, \mathcal{G}_T)}{p(Y)\, p(\mathcal{G}_S, \mathcal{G}_T)} = \log p(Y|\mathcal{G}_S, \mathcal{G}_T) - \log p(Y). \tag{24}$$

Substituting Eq. (24) into Eq. (23) yields:

$$\begin{aligned} I(Y; \mathcal{G}_S, \mathcal{G}_T) &= \mathbb{E}_{p(Y, \mathcal{G}_S, \mathcal{G}_T)}[\log p(Y|\mathcal{G}_S, \mathcal{G}_T)] - \mathbb{E}_{p(Y)}[\log p(Y)] \\ &= \mathbb{E}_{p(Y, \mathcal{G}_S, \mathcal{G}_T)}[\log p(Y|\mathcal{G}_S, \mathcal{G}_T)] + H(Y), \end{aligned} \tag{25}$$

where $H(Y) := -\mathbb{E}_{p(Y)}[\log p(Y)]$ is the entropy of $Y$, which is a constant independent of the model parameters. Taking the negative of both sides, we have:

$$-I(Y; \mathcal{G}_S, \mathcal{G}_T) = -\mathbb{E}_{p(Y, \mathcal{G}_S, \mathcal{G}_T)}[\log p(Y|\mathcal{G}_S, \mathcal{G}_T)] - H(Y). \tag{26}$$

**Variational upper bound.** The true conditional distribution $p(Y|\mathcal{G}_S, \mathcal{G}_T)$ is generally intractable in our setting. We introduce a variational classifier $q_\theta(Y|g_\phi(\mathcal{G}_S, \mathcal{G}_T))$ to approximate it, where $g_\phi$ is the fusion function that combines $(\mathcal{G}_S, \mathcal{G}_T)$ into a unified representation. By the non-negativity of KL divergence, we have:

$$\begin{aligned} 0 &\leq \mathbb{E}_{p(\mathcal{G}_S, \mathcal{G}_T)} \left[ D_{\mathrm{KL}}\big(p(Y|\mathcal{G}_S, \mathcal{G}_T) \,\|\, q_\theta(Y|g_\phi(\mathcal{G}_S, \mathcal{G}_T))\big) \right] \\ &= \mathbb{E}_{p(\mathcal{G}_S, \mathcal{G}_T)} \mathbb{E}_{p(Y|\mathcal{G}_S, \mathcal{G}_T)} \left[ \log \frac{p(Y|\mathcal{G}_S, \mathcal{G}_T)}{q_\theta(Y|g_\phi(\mathcal{G}_S, \mathcal{G}_T))} \right] \\ &= \mathbb{E}_{p(Y, \mathcal{G}_S, \mathcal{G}_T)}[\log p(Y|\mathcal{G}_S, \mathcal{G}_T)] - \mathbb{E}_{p(Y, \mathcal{G}_S, \mathcal{G}_T)}[\log q_\theta(Y|g_\phi(\mathcal{G}_S, \mathcal{G}_T))]. \end{aligned} \tag{27}$$

Rearranging the inequality, we obtain:

$$\mathbb{E}_{p(Y, \mathcal{G}_S, \mathcal{G}_T)}[\log p(Y|\mathcal{G}_S, \mathcal{G}_T)] \geq \mathbb{E}_{p(Y, \mathcal{G}_S, \mathcal{G}_T)}[\log q_\theta(Y|g_\phi(\mathcal{G}_S, \mathcal{G}_T))]. \tag{28}$$

Substituting Eq. (28) into Eq. (26) gives an upper bound on $-I(Y; \mathcal{G}_S, \mathcal{G}_T)$:

$$-I(Y; \mathcal{G}_S, \mathcal{G}_T) \leq -\mathbb{E}_{p(Y, \mathcal{G}_S, \mathcal{G}_T)}[\log q_\theta(Y|g_\phi(\mathcal{G}_S, \mathcal{G}_T))] - H(Y). \tag{29}$$

Since $H(Y)$ is constant with respect to trainable parameters, minimizing $-I(Y; \mathcal{G}_S, \mathcal{G}_T)$ is equivalent to minimizing:

$$\mathcal{L}_{\mathrm{Pre}} := -\mathbb{E}_{p(Y, \mathcal{G}_S, \mathcal{G}_T)}[\log q_\theta(Y|g_\phi(\mathcal{G}_S, \mathcal{G}_T))]. \tag{30}$$

## A.2. Compression term: variational upper bound for $I(\mathcal{G}^t_{u,v}; \mathcal{G}_S, \mathcal{G}_T)$.

We derive the bound used in Proposition 4.2 for the compression term. Recall that in practice, we work with the sampled ego-temporal subgraph $\mathcal{G}^t_{u,v}$ as the model input. For notational clarity in the derivation, we first establish the relationship for the general input graph $\mathcal{G}^t$, and then note that the same derivation applies when $\mathcal{G}^t$ is replaced by $\mathcal{G}^t_{u,v}$.

**Step 1: Markov chain reduction.** We assume the Markov chain:

$$\mathcal{G}^t \to \mathcal{G}_E \to (\mathcal{G}_S, \mathcal{G}_T), \tag{31}$$

which holds when $(\mathcal{G}_S, \mathcal{G}_T)$ are (possibly stochastic) mappings of $\mathcal{G}_E$, meaning that $(\mathcal{G}_S, \mathcal{G}_T)$ are conditionally independent of $\mathcal{G}^t$ given $\mathcal{G}_E$. By the data processing inequality, we have:

$$I(\mathcal{G}^t; \mathcal{G}_S, \mathcal{G}_T) \leq I(\mathcal{G}^t; \mathcal{G}_E). \tag{32}$$

This inequality states that the mutual information between $\mathcal{G}^t$ and $(\mathcal{G}_S, \mathcal{G}_T)$ cannot exceed the mutual information between $\mathcal{G}^t$ and the intermediate bottleneck $\mathcal{G}_E$. Thus, to upper bound $I(\mathcal{G}^t; \mathcal{G}_S, \mathcal{G}_T)$, it suffices to upper bound $I(\mathcal{G}^t; \mathcal{G}_E)$.

**Step 2: Mutual information as expected KL divergence.** By the definition of mutual information, we have:

$$
\begin{aligned}
I(\mathcal{G}^t; \mathcal{G}_E) &:= \mathbb{E}_{p(\mathcal{G}^t, \mathcal{G}_E)} \left[ \log \frac{p_\theta(\mathcal{G}_E | \mathcal{G}^t)}{p_\theta(\mathcal{G}_E)} \right] \\
&= \mathbb{E}_{p(\mathcal{G}^t)} \mathbb{E}_{p_\theta(\mathcal{G}_E | \mathcal{G}^t)} \left[ \log \frac{p_\theta(\mathcal{G}_E | \mathcal{G}^t)}{p_\theta(\mathcal{G}_E)} \right] \\
&= \mathbb{E}_{p(\mathcal{G}^t)} \left[ D_{\mathrm{KL}} \big( p_\theta(\mathcal{G}_E | \mathcal{G}^t) \,\|\, p_\theta(\mathcal{G}_E) \big) \right],
\end{aligned}
\tag{33}
$$

where the marginal distribution $p_\theta(\mathcal{G}_E) = \int p_\theta(\mathcal{G}_E | \mathcal{G}^t) \, p(\mathcal{G}^t) \, d\mathcal{G}^t$ is generally intractable because it requires integrating over all possible input graphs.

**Step 3: Variational upper bound.** To obtain a tractable upper bound, we introduce a variational marginal $q(\mathcal{G}_E)$ (a tractable prior over explanations) to approximate the intractable marginal $p_\theta(\mathcal{G}_E)$. Consider the KL divergence between $p_\theta(\mathcal{G}_E | \mathcal{G}^t)$ and $q(\mathcal{G}_E)$:

$$
\begin{aligned}
\mathbb{E}_{p(\mathcal{G}^t)} \left[ D_{\mathrm{KL}} \big( p_\theta(\mathcal{G}_E | \mathcal{G}^t) \,\|\, q(\mathcal{G}_E) \big) \right] &= \mathbb{E}_{p(\mathcal{G}^t)} \mathbb{E}_{p_\theta(\mathcal{G}_E | \mathcal{G}^t)} \left[ \log \frac{p_\theta(\mathcal{G}_E | \mathcal{G}^t)}{q(\mathcal{G}_E)} \right] \\
&= \mathbb{E}_{p(\mathcal{G}^t, \mathcal{G}_E)} \left[ \log \frac{p_\theta(\mathcal{G}_E | \mathcal{G}^t)}{q(\mathcal{G}_E)} \right] \\
&= \mathbb{E}_{p(\mathcal{G}^t, \mathcal{G}_E)} \left[ \log \frac{p_\theta(\mathcal{G}_E | \mathcal{G}^t)}{p_\theta(\mathcal{G}_E)} \right] + \mathbb{E}_{p(\mathcal{G}^t, \mathcal{G}_E)} \left[ \log \frac{p_\theta(\mathcal{G}_E)}{q(\mathcal{G}_E)} \right] \\
&= \mathbb{E}_{p(\mathcal{G}^t, \mathcal{G}_E)} \left[ \log \frac{p_\theta(\mathcal{G}_E | \mathcal{G}^t)}{p_\theta(\mathcal{G}_E)} \right] + \mathbb{E}_{p(\mathcal{G}_E)} \left[ \log \frac{p_\theta(\mathcal{G}_E)}{q(\mathcal{G}_E)} \right] \\
&= I(\mathcal{G}^t; \mathcal{G}_E) + D_{\mathrm{KL}} \big( p_\theta(\mathcal{G}_E) \,\|\, q(\mathcal{G}_E) \big) \\
&\geq I(\mathcal{G}^t; \mathcal{G}_E),
\end{aligned}
\tag{34}
$$

where the inequality follows from the non-negativity of KL divergence: $D_{\mathrm{KL}} \big( p_\theta(\mathcal{G}_E) \,\|\, q(\mathcal{G}_E) \big) \geq 0$.

**Step 4: Combining the bounds.** Combining Eq. (32) and Eq. (34), we obtain the chain of inequalities:

$$I(\mathcal{G}^t; \mathcal{G}_S, \mathcal{G}_T) \leq I(\mathcal{G}^t; \mathcal{G}_E) \leq \mathbb{E}_{p(\mathcal{G}^t)} \left[ D_{\mathrm{KL}} \big( p_\theta(\mathcal{G}_E | \mathcal{G}^t) \,\|\, q(\mathcal{G}_E) \big) \right]. \tag{35}$$

Therefore, minimizing the compression loss:

$$\mathcal{L}_{\mathrm{Com}} := \mathbb{E}_{p(\mathcal{G}^t)} \left[ D_{\mathrm{KL}} \big( p_\theta(\mathcal{G}_E | \mathcal{G}^t) \,\|\, q(\mathcal{G}_E) \big) \right] \tag{36}$$

minimizes an upper bound on $I(\mathcal{G}^t; \mathcal{G}_S, \mathcal{G}_T)$. In practice, we work with the sampled ego-temporal subgraph $\mathcal{G}_{u,v}^t$ instead of the full graph $\mathcal{G}^t$, so the expectation is taken over $p(\mathcal{G}_{u,v}^t)$ rather than $p(\mathcal{G}^t)$.

**A.3. Disentanglement term: adversarial JS proxy for $I(\mathcal{G}_S; \mathcal{G}_T | Y)$.**

We derive an adversarial proxy for the conditional mutual information $I(\mathcal{G}_S; \mathcal{G}_T | Y)$.

**Step 1: Conditional mutual information as conditional KL divergence.** By definition, the conditional mutual information between $\mathcal{G}_S$ and $\mathcal{G}_T$ given $Y$ is:

$$
\begin{aligned}
I(\mathcal{G}_S; \mathcal{G}_T | Y) &:= \mathbb{E}_{p(Y)} \left[ \mathbb{E}_{p_\theta(\mathcal{G}_S, \mathcal{G}_T | Y)} \left[ \log \frac{p_\theta(\mathcal{G}_S, \mathcal{G}_T | Y)}{p_\theta(\mathcal{G}_S | Y) \, p_\theta(\mathcal{G}_T | Y)} \right] \right] \\
&= \mathbb{E}_{p(Y)} \left[ D_{\mathrm{KL}} \big( p_\theta(\mathcal{G}_S, \mathcal{G}_T | Y) \, \| \, p_\theta(\mathcal{G}_S | Y) \, p_\theta(\mathcal{G}_T | Y) \big) \right].
\end{aligned}
\tag{37}
$$

This expresses $I(\mathcal{G}_S; \mathcal{G}_T | Y)$ as the expected KL divergence between the conditional joint distribution $p_\theta(\mathcal{G}_S, \mathcal{G}_T | Y)$ and the product of conditional marginals $p_\theta(\mathcal{G}_S | Y) \, p_\theta(\mathcal{G}_T | Y)$. Directly optimizing Eq. (37) is difficult because our model provides samples of $(\mathcal{G}_S, \mathcal{G}_T)$ but not closed-form conditional densities. We thus adopt a Jensen–Shannon (JS) divergence proxy via density-ratio estimation.

**Step 2: JS divergence variational form.** For a fixed label value $y$, we define two distributions:

$$
\begin{aligned}
P_y &:= p_\theta(\mathcal{G}_S, \mathcal{G}_T | Y = y), \\
Q_y &:= p_\theta(\mathcal{G}_S | Y = y) \, p_\theta(\mathcal{G}_T | Y = y).
\end{aligned}
\tag{38}
$$

Here, $P_y$ is the conditional joint distribution, and $Q_y$ is the product of conditional marginals. The Jensen–Shannon divergence between $P_y$ and $Q_y$ admits the following variational representation:

$$
\begin{aligned}
\mathrm{JS}(P_y \| Q_y) &= \frac{1}{2} D_{\mathrm{KL}} \left( P_y \, \Big\| \, \frac{P_y + Q_y}{2} \right) + \frac{1}{2} D_{\mathrm{KL}} \left( Q_y \, \Big\| \, \frac{P_y + Q_y}{2} \right) \\
&= \log 2 + \frac{1}{2} \max_{d_\psi} \left( \mathbb{E}_{P_y} [\log d_\psi] + \mathbb{E}_{Q_y} [\log(1 - d_\psi)] \right),
\end{aligned}
\tag{39}
$$

where $d_\psi(\cdot) \in (0, 1)$ is a discriminator function parameterized by $\psi$. The second equality follows from the variational characterization of JS divergence, where the optimal discriminator $d_\psi^*$ distinguishes between samples from $P_y$ and $Q_y$.

**Step 3: Aggregating over labels.** To obtain an objective that works across all label values, we aggregate the JS divergence over the distribution of $Y$:

$$
\begin{aligned}
\mathbb{E}_{p(Y)} [\mathrm{JS}(P_Y \| Q_Y)] &= \mathbb{E}_{p(Y)} \left[ \log 2 + \frac{1}{2} \max_{d_\psi} \left( \mathbb{E}_{P_Y} [\log d_\psi] + \mathbb{E}_{Q_Y} [\log(1 - d_\psi)] \right) \right] \\
&= \log 2 + \frac{1}{2} \max_{d_\psi} \mathbb{E}_{p(Y)} \left[ \mathbb{E}_{P_Y} [\log d_\psi] + \mathbb{E}_{Q_Y} [\log(1 - d_\psi)] \right].
\end{aligned}
\tag{40}
$$

This yields the adversarial objective used in the main text:

$$
\begin{aligned}
\mathcal{L}_{\mathrm{Dis}}(\theta, \psi) := \mathbb{E}_{p(Y)} \Big[ &\mathbb{E}_{p_\theta(\mathcal{G}_S, \mathcal{G}_T | Y)} \big[ \log d_\psi(\mathcal{G}_S, \mathcal{G}_T, Y) \big] \\
&+ \mathbb{E}_{p_\theta(\mathcal{G}_S | Y) \, p_\theta(\mathcal{G}_T | Y)} \big[ \log(1 - d_\psi(\mathcal{G}_S, \mathcal{G}_T, Y)) \big] \Big].
\end{aligned}
\tag{41}
$$

In practice, the product term $p_\theta(\mathcal{G}_S | Y) \, p_\theta(\mathcal{G}_T | Y)$ is approximated by conditional resampling: we independently resample $\tilde{\mathcal{G}}_T \sim p_\theta(\mathcal{G}_T | Y)$ under the same label $Y$ to form negative pairs $(\mathcal{G}_S, \tilde{\mathcal{G}}_T, Y)$. We update the discriminator parameters $\psi$ to maximize Eq. (41), while updating the encoder and disentangler parameters $\theta$ (and $g_\phi$ when applicable) to minimize it.

**Step 4: Connection to conditional independence.** For each label value $y$, when $\mathrm{JS}(P_y \| Q_y) = 0$, we have $P_y = Q_y$, which means:

$$
p_\theta(\mathcal{G}_S, \mathcal{G}_T | Y = y) = p_\theta(\mathcal{G}_S | Y = y) \, p_\theta(\mathcal{G}_T | Y = y).
\tag{42}
$$

This is exactly the definition of conditional independence: $\mathcal{G}_S$ and $\mathcal{G}_T$ are conditionally independent given $Y = y$. When conditional independence holds, the conditional KL divergence in Eq. (37) equals zero, hence $I(\mathcal{G}_S; \mathcal{G}_T | Y = y) = 0$. Since JS divergence is non-negative and equals zero only when the distributions are identical, minimizing $\mathcal{L}_{\mathrm{Dis}}(\theta, \psi)$ with respect to $\theta$ encourages the conditional joint to match the conditional product, thereby promoting conditional independence and making $I(\mathcal{G}_S; \mathcal{G}_T | Y)$ small.

## B. Algorithm and Complexity Analysis

We summarize the overall training procedure of ST-TGEXPLAINER in Algorithm 1. In each iteration, the model samples an explanatory bottleneck $\mathcal{G}_E$ from the input ego-temporal subgraph and disentangles it into stability and transition patterns $(\mathcal{G}_S, \mathcal{G}_T)$. The parameters are optimized under the min–max objective in Eq. (8), where the prediction and compression terms correspond to Eqs. (22) and (15) (with prior in Eq. (14)), and the disentanglement term is implemented by the adversarial discriminator loss in Eq. (19).

---

**Algorithm 1** Training procedure of ST-TGEXPLAINER

---

**Require:** Temporal event stream $\mathcal{E} = \{(u_i, v_i, t_i), Y_i\}_{i=1}^{K}$; history length $L$; prior rate $r$; weights $\beta, \gamma$.
**Require:** Learnable parameters: encoder $\theta$, predictor $\phi$, discriminator $\psi$.
 1: **while** not converged **do**
 2:     Sample a mini-batch $\mathcal{B} \subset \mathcal{E}$.
 3:     **for all** $(u, v, t), Y \in \mathcal{B}$ **do**
 4:         Construct ego-temporal subgraph $\mathcal{G}_{u,v}^t$ with $L$ most recent historical interactions.
 5:         $\mathbf{H}^t \leftarrow \text{Encoder}_\theta(\mathcal{G}_{u,v}^t)$.
 6:         Compute edge inclusion probabilities $p_e = \sigma(\text{MLP}(\mathbf{h}_e))$ and sample relaxed mask $\mathbf{m}$ (Concrete/Gumbel-Softmax).
 7:         Obtain explanatory subgraph $\mathcal{G}_E \leftarrow \mathbf{m} \odot \mathcal{G}_{u,v}^t$.
 8:         Compute frequency score $\mathbf{h}_F$ and assignment $\mathbf{p}_f = \sigma(\text{MLP}(\mathbf{h}_F))$;
 9:         $\mathcal{G}_S \leftarrow \mathbf{p}_f \odot \mathcal{G}_E, \quad \mathcal{G}_T \leftarrow (1 - \mathbf{p}_f) \odot \mathcal{G}_E$.
10:         $\mathbf{h}_S \leftarrow \text{Mean}(\text{Encoder}_\theta(\mathcal{G}_S)), \quad \mathbf{h}_T \leftarrow \text{Mean}(\text{Encoder}_\theta(\mathcal{G}_T))$.
11:         Fuse $\mathbf{h}_E \leftarrow g_\phi(\mathbf{h}_S, \mathbf{h}_T)$ and predict $\hat{Y}$.
12:         Compute $\mathcal{L}_{\text{Pre}}$ via Eq. (22).
13:         Compute $\mathcal{L}_{\text{Com}}$ via Eq. (15) (with prior $q(\mathcal{G}_E)$ in Eq. (14)).
14:         Compute $\mathcal{L}_{\text{Dis}}$ via Eq. (19) (negative pairs by conditional resampling).
15:     **end for**
16:     Update discriminator: $\psi \leftarrow \psi + \eta_\psi \nabla_\psi \mathcal{L}_{\text{Dis}}$            $\triangleright$ maximize
17:     Update model: $(\theta, \phi) \leftarrow (\theta, \phi) - \eta \nabla_{\theta,\phi}(\mathcal{L}_{\text{Pre}} + \beta \mathcal{L}_{\text{Com}} + \gamma \mathcal{L}_{\text{Dis}})$     $\triangleright$ minimize Eq. (8)
18: **end while**

---

**Complexity analysis.** Let $L$ denote the number of sampled neighbors (history length), $d$ the embedding dimension, and $E$ the number of interactions/edges in the dataset. Our encoder is an MLP-Mixer operating on an input tensor of shape $L \times d$. Within each encoder layer, token mixing applies an MLP along the token dimension for each channel, which costs $O(L^2 d)$, and channel mixing applies an MLP along the channel dimension for each token, which costs $O(L d^2)$. Therefore, the per-layer encoder complexity is $O(L^2 d + L d^2)$. The frequency-guided assignment in Eq. (16) is a lightweight MLP evaluated over the $L$ historical edges, adding $O(Ld)$ computation, which is lower-order compared to the encoder. Overall, the per-interaction complexity is dominated by $O(L^2 d + L d^2)$ up to constant factors from a constant number of encoder invocations in the pipeline. Across the whole dataset, the total complexity scales as $O(E(L^2 d + L d^2))$.

## C. Dataset Details

We briefly introduce the datasets used in our experiments:

- **Wikipedia**[1] consists of 9,227 nodes (editors and Wikipedia pages) and 157,474 edges representing timestamped edit requests. Each edge is associated with a 172-dimensional Linguistic Inquiry and Word Count (LIWC) feature vector of the requested text.

- **Reddit**[2] consists of a bipartite graph monitoring user posts on Reddit for one month. Nodes represent users and subreddits, while edges denote timestamped posting requests, each accompanied by a 172-dimensional LIWC feature. This dataset also includes dynamic labels indicating whether users were banned from posting.

---

[1] http://snap.stanford.edu/jodie/wikipedia.csv
[2] http://snap.stanford.edu/jodie/reddit.csv

- **UCI**[3] is a social network within the online community of University of California, Irvine students spanning 196 days. Nodes represent students, and edges denote messages exchanged between two students, each with a timestamp in seconds.

- **Enron**[4] is an email communication network consisting of timestamped emails exchanged within the Enron corporation over a period of three years.

- **USLegis**[5] is a co-sponsorship network among U.S. senators. Each node represents a legislator, and two legislators are connected if they have jointly sponsored a bill. The weight assigned to each edge represents the number of times two legislators have jointly sponsored bills over 12 terms.

- **Can.Parl.**[6] captures interactions among Canadian Members of Parliament from 2006 to 2019. Nodes represent Members of Parliament, and two members are connected if they vote the same way on a bill. The weight assigned to each edge represents the number of bills on which the two members cast the same vote within one year.

## D. Baseline Details

Details on the compared baselines for each experiment are provided below:

### D.1. Link Prediction

- **JODIE** (Kumar et al., 2019) utilizes two interconnected recurrent neural networks to update user and item states, enhancing precision in capturing intricate temporal patterns within user-item interactions. The incorporation of a projection operation enables accurate learning of future representation trajectories.

- **DyRep** (Trivedi et al., 2019) introduces a recurrent architecture that systematically updates node states after each interaction within temporal graphs. Furthermore, it incorporates a temporal-attentive aggregation module, enabling the model to consider the evolving structural information in temporal graphs over time. This dual-component framework provides a consistent and effective approach to capture intricate temporal dynamics and evolving graph structures in dynamic networks.

- **TGAT** (Xu et al., 2020) utilizes self-attention to compute node representations, aggregating features from temporal neighbors and employing a time encoding function for capturing nuanced temporal patterns. This concise approach ensures precise analysis of evolving graph structures and temporal dynamics in dynamic networks.

- **TGN** (Rossi et al., 2020) utilizes an evolving memory system for each node, updated upon node interactions through the message function, message aggregator, and memory updater mechanisms. Simultaneously, an embedding module is employed to generate temporal node representations, ensuring a dynamic and comprehensive analysis of evolving graph structures in complex networks.

- **CAWN** (Wang et al., 2021) initiates its process by extracting multiple causal anonymous walks for each node, facilitating an in-depth examination of the network dynamics' causality and the establishment of relative node identities. Following this, RNN is utilized to encode each individual walk. The encoded walks are subsequently aggregated to synthesize the final node representation.

- **GraphMixer** (Cong et al., 2023) adopts a fixed time encoding function, surpassing its trainable counterpart in performance. It incorporates this function into an MLP-Mixer-based link encoder to learn from temporal links efficiently. The framework utilizes neighbor mean-pooling in the node encoder for a concise summarization of node features.

- **DyGFormer** (Yu et al., 2023) introduces a Transformer-based architecture enhanced by a neighbor co-occurrence coding scheme, effectively capturing node correlations within interactions. Also, it employs patching techniques to enable the model to capture long-term temporal dependencies.

---

[3] http://konect.cc/networks/opsahl-ucforum/
[4] https://www.cs.cmu.edu/~enron/
[5] https://zenodo.org/records/7213796/files/USLegis.zip?download=1
[6] https://zenodo.org/records/7213796/files/CanParl.zip?download=1

- **FreeDyG** (Tian et al., 2024) incorporates a node interaction frequency encoding module that captures the proportion of common neighbors and the frequency of interactions between node pairs. Shifting the focus to the frequency domain allows the model to better capture periodic patterns and dynamic shifts in node interactions over time.

## D.2. Explanation

- **Grad-CAM** (Pope et al., 2019) provides post-hoc explanations by using importance scores computed from the gradients of the logit with respect to node embeddings.

- **GNNExplainer** (Ying et al., 2019) is a post-hoc method that provides explanations for GNN predictions. Specifically, it learns to mask the input graph while incorporating label information into the selected subgraphs.

- **PGExplainer** (Luo et al., 2020) uses an explanation network to parameterize edge distributions and construct an explanation subgraph. The explanation network is optimized by maximizing the mutual information between the explanatory subgraph and the GNN's predictions.

- **T-GNNExplainer** (Xia et al., 2023) aims to explain TGNNs post-hoc, employing both navigator and explorer components. The pretrained navigator captures inductive relationships among events, and the explorer seeks the optimal combination of candidates for explanation.

- **TempME** (Chen & Ying, 2023) is a post-hoc explanation method for temporal graphs. It follows a motif-based approach and is only partially comparable with our event-level explanation framework.

- **CoDy** (Qu et al., 2025) is a counterfactual explainer that leverages Monte Carlo Tree Search to efficiently identify minimal event subsets whose removal flips the TGNN prediction.

- **TGIB** (Seo et al., 2024) is a self-explainable temporal graph neural network based on the information bottleneck principle. It integrates prediction and explanation by incorporating stochastic masks into temporal events, allowing the model to identify the most informative interactions during training.

# E. Detailed Experimental Settings

## E.1. Link Prediction Task Setting

For the link prediction task, we utilize Average Precision (AP) and Mean Reciprocal Rank (MRR) as evaluation metrics, which are widely recognized in the literature (Yi et al., 2025). For AP, we randomly sample one negative instance from all potential destinations, while for MRR, we randomly sample 100 negative destinations. Following the TGB-Seq ranking protocol (Yi et al., 2025), for each test interaction $(u, v, t)$, we rank the true destination $v$ among a candidate set consisting of $v$ plus the sampled negative destinations (excluding $v$), without filtered evaluation. The dataset is split chronologically into 70%/15%/15% for train/validation/test to avoid future leakage.

## E.2. Explainability Task Setting

The fidelity metric used in T-GNNExplainer, which serves as an evaluation criterion, has a range that can vary significantly depending on the dataset or backbone model. As a result, computing areas under the curves associated with fidelity scores may not yield consistent or comparable evaluation outcomes across different settings. To address this limitation and provide a more reliable assessment of explanation quality, we instead measure the proportion of cases where the predictions made on explanation subgraphs remain consistent with those of the original model. To quantitatively evaluate the faithfulness of explanation subgraphs, we adopt the ACC–AUC metric. Following (Chen & Ying, 2023), this metric assesses how well predictions made by explanation subgraphs align with the original model's predictions over varying levels of sparsity. Specifically, let $f(\cdot)$ denote the trained base model, and let $\hat{y}_i(\mathcal{G}^t)$ be its prediction for the $i$-th instance on the full graph $\mathcal{G}^t$. For a given sparsity level $s \in [0, 0.3]$, an explanation subgraph $\mathcal{G}_{\mathrm{E}}^{(s)}$ is constructed by retaining the top-$s$ proportion of explanatory edges determined by the explainer. The prediction accuracy at each sparsity level $s$ is then defined as

$$\mathrm{ACC}(s) \ = \ \frac{1}{N} \sum_{i=1}^{N} \mathbf{1}\Big(\hat{y}_i(\mathcal{G}^t) = \hat{y}_i\big(\mathcal{G}_{\mathrm{E}}^{(s)}\big)\Big), \tag{43}$$

where $N$ denotes the number of evaluated instances and $\mathbf{1}(\cdot)$ is the indicator function. To obtain a single scalar score that summarizes the explanation performance across different sparsity levels, ACC–AUC is computed as the area under the accuracy–sparsity curve:

$$\text{ACC-AUC} \;=\; \sum_{j=1}^{m-1} \text{ACC}(s_j)\left(s_{j+1} - s_j\right), \tag{44}$$

where $\{s_j\}_{j=1}^{m}$ denotes the discretized sparsity levels sampled uniformly from $0$ to $0.3$ (e.g., with a step size of $0.002$). Intuitively, ACC–AUC quantifies the proportion of explanation subgraphs that preserve the original model predictions over a wide range of sparsity constraints. A higher ACC–AUC value indicates that the extracted explanations are both faithful and robust. Compared to the fidelity metric adopted in T-GNNExplainer, which may exhibit substantial variance across datasets and backbone models, ACC–AUC provides a more stable and interpretable evaluation by integrating prediction consistency across sparsity levels.

### E.3. Experimental Setup

We closely follow (Yu et al., 2023; Yi et al., 2025) in our dataset preparation, chronologically splitting the data with a ratio of 70%, 15%, and 15% for training, validation, and testing, respectively. All models undergo a training regimen of up to 100 epochs, incorporating an early stopping strategy with a patience of 10. The model achieving optimal performance on the validation set is selected for subsequent testing. The Adam optimizer is uniformly employed across all models, utilizing supervised binary cross-entropy loss as the objective function, with a consistent learning rate of 0.0001 and a batch size of 200. Performance is evaluated as the average over 5 independent runs with different random seeds. The baseline implementation is based on the DyGLib library (Yu et al., 2023).

We conduct our experiments on a machine with an Intel(R) Xeon(R) Gold 6330 CPU @ 2.00GHz, 256 GiB RAM, and four NVIDIA 3090 GPUs. All methods are implemented in Python 3.9 with PyTorch.

*Table 6.* The tuning ranges of hyperparameters for ST-TGEXPLAINER

| Hyperparameter | Tuning Range | Description |
|---|---|---|
| $L$ | [20, 30, 40, 50, 60] | history length / number of sampled historical interactions |
| $\beta$ | [0.1, 0.2, 0.4, 0.6, 0.8, 1.0] | the balance weight of loss $\mathcal{L}_{\text{Com}}$ |
| $\gamma$ | [0.1, 0.2, 0.4, 0.6, 0.8, 1.0] | the balance weight of loss $\mathcal{L}_{\text{Dis}}$ |
| $N_{\text{layers}}$ | [1, 2] | the number of layers in MLP-Mixer |
| Dropout | [0.1, 0.2, 0.3, 0.4, 0.5] | the dropout ratio for model |

### E.4. Hyperparameters

For the baselines on the TGB-Seq benchmark, the TGB benchmark, and six commonly used datasets (Wikipedia, Reddit, UCI, Enron, USLegis, and Can.Parl.), we adopt the best hyperparameter configurations reported in (Yu et al., 2023; Huang et al., 2023; Yi et al., 2025), respectively. Owing to ST-TGEXPLAINER's simple architecture, it involves only a small set of tunable hyperparameters. We fix the batch size across all methods and set the embedding dimension to 172 for all datasets. For the prior parameter $r$ in Eq. (14), we use $r = 0.7$ on all datasets. Inspired by curriculum learning (Bengio et al., 2009), we initialize $r$ at 0.9 and apply a step decay of 0.1 every 10 epochs until reaching the tuned value. The temperature in the Gumbel–Softmax trick (Jang et al., 2017) is not tuned and is fixed to 1 across all datasets. Table 6 summarizes the tuning ranges, and we select the final configuration via grid search on the validation set.

## F. Additional Experimental Results

### F.1. Explanation Performance

To directly evaluate explanation fidelity beyond prediction consistency, we adopt **AUFSC** (Area Under the Fidelity–Sparsity Curve) from the CoDy protocol (Qu et al., 2025). AUFSC integrates fidelity across sparsity levels into a single $[0, 1]$ score, with two complementary views: (i) $\text{AUFSC}^+$ (necessity): removing the identified important edges should change the prediction; and (ii) $\text{AUFSC}^-$ (sufficiency): keeping *only* the identified important edges should preserve the prediction. We further report AUFSC separately on correctly and incorrectly predicted instances.

*Table 7.* Explanation results for the AUFSC$^+$ and AUFSC$^-$ of different explanation methods applied to the GraphMixer model. Results are reported for UCI and Wikipedia datasets.

| Method | AUFSC$^+$ | | | | AUFSC$^-$ | | | |
| --- | --- | --- | --- | --- | --- | --- | --- | --- |
| | Correct | | Incorrect | | Correct | | Incorrect | |
| | Wikipedia | UCI | Wikipedia | UCI | Wikipedia | UCI | Wikipedia | UCI |
| PGExplainer | 0.03 | 0.02 | 0.11 | 0.08 | 0.67 | 0.39 | 0.54 | 0.61 |
| T-GNNExplainer | 0.01 | 0.05 | 0.10 | 0.14 | 0.61 | 0.45 | 0.43 | 0.53 |
| TempME | 0.04 | 0.06 | 0.15 | 0.22 | 0.72 | 0.56 | 0.50 | 0.84 |
| TGIB | 0.04 | 0.04 | 0.18 | 0.20 | 0.69 | 0.49 | 0.41 | 0.78 |
| CoDy | 0.15 | 0.16 | 0.21 | 0.39 | 0.82 | 0.65 | 0.54 | 0.87 |
| Ours | 0.15 | 0.22 | 0.24 | 0.39 | 0.91 | 0.86 | 0.62 | 0.91 |

As shown in Table 7, ST-TGEXPLAINER achieves the best or tied-best AUFSC$^+$ and substantially improves AUFSC$^-$ (e.g., UCI Correct: 0.86 vs. CoDy 0.65), indicating that the extracted explanations capture evidence that is both necessary and sufficient for the model's predictions.

## F.2. Complexity Performance

See Figure 7 for details of the UCI dataset results with respect to AP, training time, and parameter size.

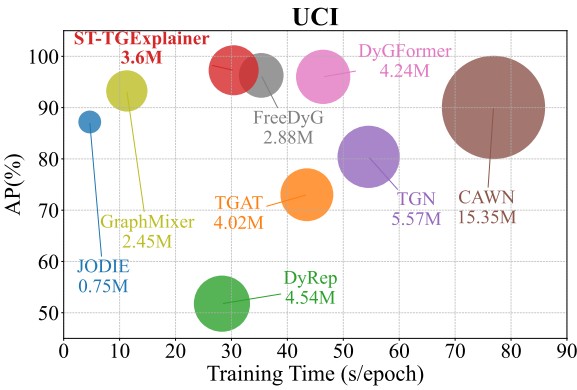

*Figure 7.* Comparison of model performance, parameter size, and training time per epoch on UCI.

## F.3. Ablation Study

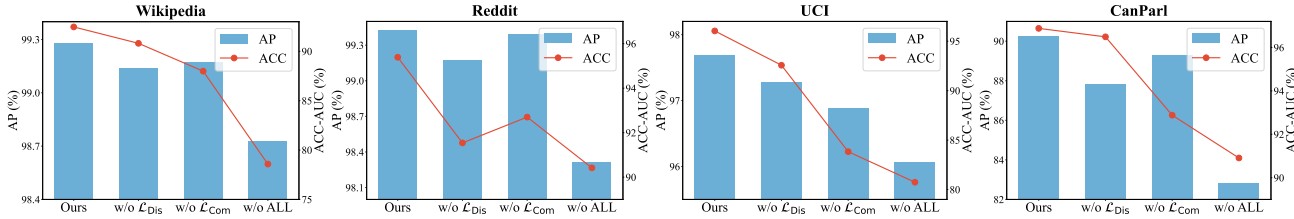

*Figure 8.* Ablation study link prediction AP (%) and explanation ACC-AUC (%) values on another four datasets.

Figure 8 extends the component ablations to four additional datasets (Wikipedia, Reddit, UCI, and Can.Parl.), showing that the prediction, compression, and disentanglement losses are consistently beneficial.

## F.4. Visualization

We also validate the effectiveness of our disentanglement approach on four additional datasets (Reddit, Enron, UCI, and Can.Parl.). Figure 9 presents PCA projections of the learned latent vectors $\mathbf{h}_S$, $\mathbf{h}_T$, and $\mathbf{h}_E$ in the feature space, where $\mathbf{h}_E$

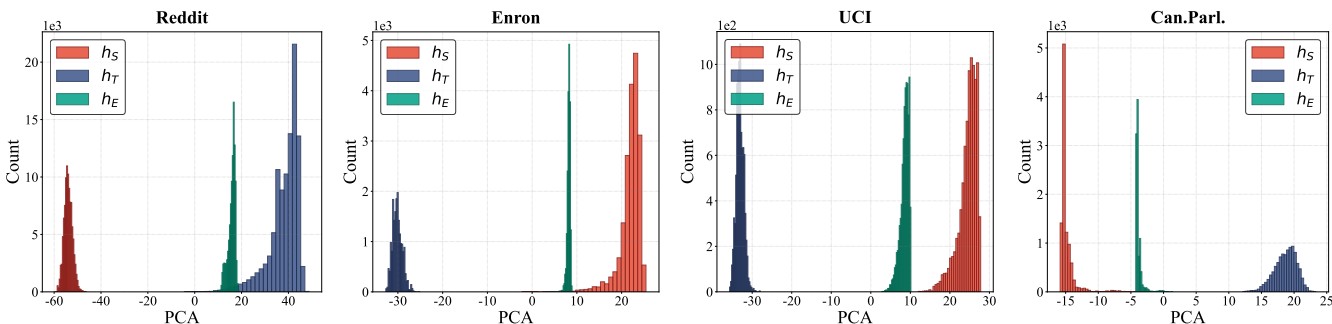

*Figure 9.* Visualization of the embedding distributions for the stability graph ($\mathbf{h}_S$), transition graph ($\mathbf{h}_T$), and explanatory graph without disentanglement ($\mathbf{h}_E$).

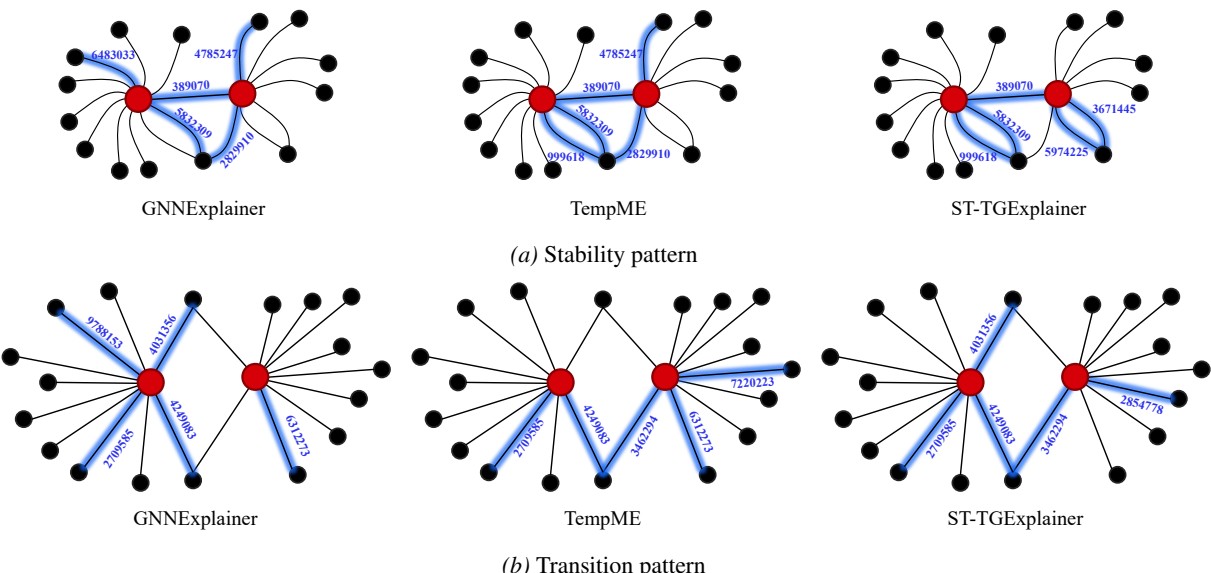

*(a)* Stability pattern

*(b)* Transition pattern

*Figure 10.* Comparison of explanation visualizations across baselines and ST-TGEXPLAINER.

denotes the representation learned from the explanatory graph without the stability–transition split.

### F.5. Case study of Explanations

Figure 10 compares visual explanations produced by the static GNNExplainer and the temporal explainer TempME with those of ST-TGEXPLAINER, for both stability and transition interactions. Red nodes denote the endpoints of the target event, blue solid edges indicate selected explanatory events, and the annotated values report the time gaps between the target and each explanatory event. Since we use GraphMixer as the base model, all explanation subgraphs are restricted to the one-hop neighborhood of the target node pair. Two observations follow: (1) GNNExplainer tends to select events that are temporally distant from the target, suggesting that static explainers transfer poorly to temporal graphs; and (2) in Figure 10(a,b), ST-TGEXPLAINER more clearly disentangles stability and transition evidence, yielding cleaner, pattern-specific explanations. Compared with TempME, which prioritizes cohesive explanation, ST-TGEXPLAINER emphasizes the separation of distinct evolution patterns.

