# OpenReview forum: "ST-TGExplainer: Disentangling Stability and Transition Patterns for Temporal GNN Interpretability"
_ICML.cc/2026/Conference — ICML 2026 regular_

### Official Review · Reviewer_X78n · 2026-02-17

**Soundness:** 3
**Presentation:** 2
**Significance:** 2
**Originality:** 3
**Overall Recommendation:** 4
**Confidence:** 3

**Summary:**

This paper introduces a temporal GNN explainer, this explainer splits historical evidence into stability graph and transition graph interactions for temporal link prediction, it learns a compact explanation subgraph and combines the two pattern representations for prediction.

**Compliance With Llm Reviewing Policy:**

Affirmed.

**Final Justification:**

My concerns have been addressed.

**Key Questions For Authors:**

Refer to ''Strengths And Weaknesses'' section.

**Limitations:**

The authors should add a section discussing the limitations of the proposed method.

**Strengths And Weaknesses:**

Strengths:
1. The paper is well organized and clearly written.

2. The proposed method focuses on a gap in temporal link prediction.

Weaknesses:

1. In the introduction, the authors use MRR to illustrate the gap between seen and unseen interactions, but the evaluation details are insufficient. It would be helpful to introduce how MRR is used in this setting and clarify the ranking protocol. This should include how the candidate set is constructed, whether a filtered evaluation is adopted, and how negative samples are generated. If the authors decide to use the MRR gap as evidence of stability bias,they need to explain more explicitly how degraded ranking performance on unseen interactions translates into biased explanatory evidence.

2. The proposed method introduces multiple changes, making it difficult to attribute the improvement specifically to the transition graph itself. It would be helpful to add other ablation studies: evaluate (1) baseline with G_S only, (2) baseline with G_T only, and (3) baseline that uses G_E, which would more directly support the claim.

3. If different explainers are evaluated with different backbones, post-hoc explainers using GraphMixer as the backbone while self-explainable methods use their own trained predictors, the improvement reported in Table 3 may partially from a stronger base model rather than a genuinely better explanation mechanism. To enhance the fairness and interpretability of comparison, authors may adopt the following measures: (1) fix a single trained TGNN as a shared backbone and evaluate all explainers on it, or (2) repeat the explainability evaluation across multiple backbones and report averaged results.

4. The standard deviations reported in Tables 2 and 3 are relatively large for some datasets, making it difficult to evaluate the stability of the results. The authors could explain the reasons for this, specify the number of runs, and clarify whether negative samples and data partitions were fixed across different runs. For some cases with very small gaps, they could add the statistical significance test.

---

> ### Author Rebuttal · Authors · 2026-03-31
>
> We thank the reviewer for their thoughtful feedback. Below, we address the raised concerns in detail.
>
> **W1: MRR Evaluation Protocol and Stability Bias**
>
> We follow the TGB-Seq[1] ranking protocol (Appendix E): for each test interaction $(u,v,t)$, we rank the true destination v among a candidate set consisting of $v$ plus 100 uniformly sampled negative destinations from the node set (excluding $v$). Data is split chronologically (70/15/15) with no future leakage. We do not use filtered evaluation, consistent with the benchmark setting.
>
> Fig.1 shows a consistent MRR drop on unseen (first-time) node pairs, suggesting weaker generalization when pair-level recurrence evidence is absent. This matters for explanation because post-hoc explainers attribute importance over the predictor’s available historical evidence/representations; if unseen-case decisions rely more on recurrent structure/neighborhood proxies, explanations may tend to emphasize recurrent (stability) evidence and under-represent emerging signals. This motivates explicitly modeling and disentangling stability vs. transition evidence during training; we will clarify this link in the revised introduction/caption.
>
>
> **W2: Additional Ablation Study**
>
> We conduct the requested ablation on Wikipedia and UCI:
>
> |Variant|Wikipedia AP/ACC-AUC|UCI AP/ACC-AUC|
> |-|-|-|
> |Fullmodel|99.21/92.47|97.28/96.04|
> |$G_E$-only|98.14/88.58|96.06/88.06|
> |$G_S$-only|98.01/87.93|91.76/84.67|
> |$G_T$-only|96.36/77.61|93.62/85.18|
>
> The results help isolate the contribution of the stability–transition decomposition: $G_S$-only is stronger on Wikipedia (repeat ratio 88.41%), while $G_T$-only is relatively stronger on UCI (repeat ratio 66.06%). The full model surpasses $G_E$-only by **+3.89 (Wikipedia)** and **+7.98 (UCI)** ACC-AUC, indicating that the stability–transition split provides complementary evidence beyond using a single mixed subgraph.
>
>
> **W3: Enhancing fairness and interpretability of the explanation comparison**
>
> Different backbone predictors can affect explanation evaluation. To further strengthen the fairness and interpretability of our comparison, we add both measures suggested by the reviewer.
> (1) Shared-backbone evaluation (fix one trained TGNN).
> For all post-hoc explainers, we fix a single trained predictor as the shared backbone. Specifically, we use the trained MLP-Mixer encoder described in our paper together with its prediction head, and we apply each model-agnostic explainer to this frozen backbone under the same ACC-AUC protocol. Results (ACC-AUC, mean±std over 5 runs) on Wikipedia/UCI are:
>
> |Models|Wikipedia|UCI|
> |-|-|-|
> |GNNExplainer|87.21±0.73|63.02±0.77|
> |PGExplainer|86.19±1.34|62.76±1.36|
> |T-GNNExplainer|89.69±0.96|82.47±0.87|
> |TempME|89.15±0.40|90.87±0.52|
> |TGIB|88.09±0.68|87.06±1.04|
> |Ours|92.47±0.26|96.04±0.39|
>
> Under this shared-backbone setting, our method remains best, suggesting that the improvements are not solely attributable to a stronger base predictor.
>
> (2) Multi-backbone evaluation (repeat across backbones).
> To further address backbone sensitivity, we additionally evaluate post-hoc explainers across three commonly used backbones (TGAT, TGN, GraphMixer) and report the averaged ACC-AUC. The averaged ACC-AUC (%) is:
>
> |Method|Wikipedia|Reddit|UCI|Enron|USLegis|Can.Parl.|
> |-|-|-|-|-|-|-|
> |GNNExplainer|89.72|93.35|72.695|80.22|87.33|85.40|
> |PGExplainer|87.91|93.06|71.879|75.379|91.47|85.00|
> |TGNNExplainer|88.98|93.92|80.99|84.74|91.27|86.38|
> |TempME|90.59|94.80|86.62|84.73|95.16|90.36|
> |Ours|92.47|95.39|96.04|90.15|97.49|96.98|
>
> Across datasets, our method remains the top performer under both shared-backbone and multi-backbone evaluations. For self-explainable methods (ours and TGIB), we keep the standard setting where the encoder and explainer are jointly optimized, consistent with the original experimental protocol.
>
>
> **W4: Standard Deviations and Statistical Significance**
>
> All results are averaged over 5 runs with different seeds; std is computed over these 5 runs. The chronological split (70/15/15) is fixed; negative samples are re-sampled per run (Appendix E.3). Variance is typically small on high-repeat datasets and larger on low-repeat/harder datasets across methods, while our performance gaps generally exceed combined std. We run two-tailed Welch’s t-tests (n=5) for the smallest-gap comparisons:
>
> |Comparison|Gap|t-stat|p-value|
> |-|-|-|-|
> |UCI AP : Ours vs. FreeDyG|+1.00|4.29|<0.02|
> |Enron AP : Ours vs. DyGFormer|+2.40|8.91|<0.001|
> |Wiki ACC-AUC : Ours vs. TempME|+2.32|13.06|<0.001|
>
> All improvements are statistically significant ($p$<0.05). We will include these tests in the revised Tables 2 and 3.
>
> **W5: Limitations Discussion**
>
> We already discuss limitations in Sec.6 (one-hop history; heuristic frequency-based disentangler), and we will expand and make this discussion more prominent in the revision.
>
> [1] TGB-Seq benchmark: Challenging temporal gnns with complex sequential dynamics. ICLR 2025.

---

> > ### Author Rebuttal · Reviewer_X78n · 2026-04-02
> >
> > Thank you for your detailed response. I have updated my score.

---

> > > ### Author Response · Authors · 2026-04-03
> > >
> > > Thank you for your time and effort in reviewing our manuscript and for the follow-up. We appreciate that you have updated your score after considering our rebuttal. We will incorporate the clarifications and additional results discussed in the response into the final version, and further refine the presentation accordingly. Thank you again for your careful and constructive review.

---

### Official Review · Reviewer_pp5Q · 2026-03-09

**Soundness:** 2
**Presentation:** 2
**Significance:** 2
**Originality:** 2
**Overall Recommendation:** 4
**Confidence:** 4

**Summary:**

This paper proposes ST-TGEXPLAINER, a self-explainable temporal graph neural network designed to enhance interpretability by disentangling historical interactions into stability and transition patterns. By employing a disentangled information bottleneck objective, the model effectively separates recurring behavioral patterns from newly emerging interactions, ensuring that frequent stable signals do not overshadow informative transition evidence. Through adversarial training and variational inference, they learn a compact explanatory subgraph that maintains high predictive accuracy while providing more faithful explanations for dynamic events.

**Compliance With Llm Reviewing Policy:**

Affirmed.

**Final Justification:**

The authors addressed my concerns well, and I will change my score to score 4.

**Key Questions For Authors:**

Please refer to the above.

**Limitations:**

Yes

**Strengths And Weaknesses:**

Strengths
- ST-TGEXPLAINER is the first to explicitly separate Stability and Transition. This prevents stability bias, where common routine actions drown out the important signals from rare but critical new events.
- The model is built on a solid mathematical foundation called the Disentangled Information Bottleneck (DIB).
- Through extensive experiments, the paper demonstrates that the model not only predicts well but also generates more faithful explanations.

Weakness
- Explain examples where stability and transition patterns exert significant influence in specific temporal graph datasets. This is a crucial process for justifying the motivation of this paper.
- There is no explicit process to induce G_S and G_R into stability and transition patterns during the training phase. Separating the two patterns through adversarial learning alone cannot guarantee that G_S ensures stability and G_R captures transition.
- How were the "newly emerging interactions" in Figure 1 defined? As this serves as the motivating experiment, a detailed explanation is required.
- Execution time experiments and complexity analysis are missing.
- The source code has not been made public.

---

> ### Author Rebuttal · Authors · 2026-03-31
>
> We would like to express our sincere appreciation to the reviewer for providing us with detailed suggestions. We have carefully reviewed each comment and offer the following responses.
>
> **W1: Stability and Transition Patterns in Datasets**
>
> We clarify with concrete dataset characteristics as shown in Table 1. Wikipedia has a high repeat ratio (88.41%), meaning many interactions recur between the same node pairs, so explanations can be dominated by recurrent (stability) evidence. In contrast, lower repeat ratio datasets such as Can.Parl. (31.08%) and USLegis (56.25%) contain a substantially larger fraction of non-recurrent / newly observed pairs, where relying only on recurrent evidence is insufficient. This is consistent with Fig.1: all evaluated TGNNs show a clear MRR drop on unseen (first-time) pairs, and the gap is larger on datasets with higher unseen ratios. Our design explicitly models both $G_S$ and $G_T$ so that emerging evidence is not overwhelmed by recurrent signals, which is also reflected by our largest AP gains over TGIB on low-repeat datasets (Can.Parl. +3.18, USLegis +2.86, UCI +3.68).
>
>
> **W2: Semantic Grounding of $G_S$ and $G_T$**
>
> Adversarial learning alone is insufficient to ensure semantic alignment, and our method is not based on adversarial disentanglement alone. Our method includes an explicit inductive bias: the frequency-guided assignment (Eq.16–17) computes co-occurrence counts $h_F$ and maps them to $p_f=\sigma(\mathrm{MLP}(h_F))$, assigning higher $p_f$ to recurrent edges (thus contributing more to $G_S$) and lower $p_f$ to weakly supported edges (thus contributing more to $G_T$). On top of this, $L_{Dis}$ (Eq.19) suppresses label-conditioned redundancy between $h_S$ and $h_T$. The “w/o $L_{Dis}$” ablation in Fig.4 shows degraded AP and ACC-AUC when removing this term, suggesting that $L_{Dis}$ refines the separation induced by the frequency prior rather than being the sole driver. (We interpret $G_R$ in the review as our $G_T$.)
>
> **W3: Definition of "Newly Emerging Interactions" in Figure 1**
>
> Following the TGB-Seq[1] benchmark protocol, an interaction $(u,v,t)$ is "seen" if $(u,v)$ has appeared at least once before $t$, and "unseen" (newly emerging) otherwise. Statistics are:
>
> |Dataset|\#Test|\#Seen|\#Unseen|Unseen%|
> |:-|:-|:-|:-|:-|
> |Wikipedia|23,621|21,077|2,544|10.77%|
> |Reddit|100,867|93,873|6,994|6.93%|
> |UCI|8,976|6,399|2,577|28.71%|
> |Enron|18,785|16,732|2,053|10.93%|
> |USLegis|4,950|3,268|1,682|33.98%|
> |Can.Parl.|10,113|4,615|5,498|54.37%|
>
> In test edge unseen ratios range from 6.93% to 54.37%, which helps interpret why the MRR gap in Fig.1 is larger on datasets with more unseen pairs. We will add this definition and statistics to the revised caption.
>
> **W4: Runtime and Complexity**
>
> We add explanation generation time (seconds per explanation) on Wikipedia/UCI,
>
> |Method|Wikipedia|UCI|
> |-|-|-|
> |T-GNNExplainer|68.6|68.5|
> |TempME|75.4|88.4|
> |TGIB|120.1|147.5|
> |CoDy|105.0|115.8|
> |Ours|53.1|66.7|
>
> Our method achieves the fastest or near-fastest runtime, notably ~2.26× faster than TGIB on Wikipedia. For link prediction, Figure 3 reports AP, training time per epoch, and parameter size, showing competitive accuracy with a compact footprint.
>
> *Complexity*: Let $L$ denote the number of sampled neighbors, $d$ the embedding dimension, and $E$ the number of interactions/edges in the dataset. Our encoder is an MLP-Mixer operating on an input tensor of shape $L \times d$. Within each encoder layer, token mixing applies an MLP along the token dimension for each channel, which costs $O(L^2 d)$, and channel mixing applies an MLP along the channel dimension for each token, which costs $O(L d^2)$. Therefore, the per-layer encoder complexity is $O\left(L^2 d + L d^2\right)$. The frequency-guided assignment in Eq.(16--17) is a lightweight MLP evaluated over the $L$ historical edges, adding $O(Ld)$ computation, which is lower-order compared to the encoder. Overall, the per-interaction complexity is dominated by $O\left(L^2 d + L d^2\right)$, up to constant factors from a constant number of encoder invocations in the pipeline. Across the whole dataset, the total complexity scales as $O\left(E(L^2 d + L d^2)\right)$.
>
> **W5: Code Availability**
>
> The anonymous repository is available at: https://anonymous.4open.science/r/ST-TGExplainer
>
> [1] TGB-Seq benchmark: Challenging temporal gnns with complex sequential dynamics. ICLR 2025.

---

> > ### Author Rebuttal · Reviewer_pp5Q · 2026-04-04
> >
> > The authors addressed my concerns well, and I will change my score to score 4.

---

> > > ### Author Response · Authors · 2026-04-04
> > >
> > > Thank you very much for your positive feedback and for raising the score. We are truly glad that our clarification addressed your concerns. We will make sure to include this discussion in the final manuscript. We sincerely appreciate your time and effort.

---

### Official Review · Reviewer_27S1 · 2026-03-10

**Soundness:** 3
**Presentation:** 4
**Significance:** 3
**Originality:** 3
**Overall Recommendation:** 5
**Confidence:** 4

**Summary:**

This work focuses on the interpretability problem of TGNNs and points out that the challenge of existing methods lies in the fact that they only focus on seen historical interactions but ignore newly emerging first-time interactions. This work proposes ST-TGExplainer, a self-explainable TGNN model, to capture two patterns, i.e., stability patterns and transition patterns—for better temporal GNN explainability. The experiments in this work could verify the effectiveness of the proposed method and demonstrate the quality of the explanations.

**Compliance With Llm Reviewing Policy:**

Affirmed.

**Final Justification:**

The rebuttal adequately addressed my questions regarding explanation metrics, encoder architecture, and the decomposition of unseen interactions. I maintain my positive assessment and am satisfied with the overall quality of the work.

**Key Questions For Authors:**

Please refer to the Weakness.

**Limitations:**

Yes

**Strengths And Weaknesses:**

## Strength
S1: Novelty. This work specially decomposes the interpretability problem in TGNNs into two patterns, stability patterns and transition patterns, and clearly demonstrates the motivations and the two-fold challenges. This is novel to me, especially identifying the first-time interactions for discovering the unseen in history.

S2: Methodology. The overall method design of the proposed ST-TGExplainer is rational to me. The method contains modular components, covering (1) an informative interaction selector; (2) a stability–transition graph disentangler; and (3) an explainable pattern feature ensembler. In addition, the disentangled Information Bottleneck analysis and propositions on temporal graphs are clear and appear to help extract an explanatory subgraph.

S3: Paper Quality. This paper is well-organized, with a clear structure, and the language is not hard to follow. The experiments are demonstrated well, with comprehensive results and visualizations.
## Weakness
I have some questions here.

Q1: Could the authors clarify what explanation sparsity means in this work? and why ACC-AUC over different sparsity levels is the metric of explanation quality? I think ACC-AUC is still derived from downstream predictive accuracy, so whether this metric may purely reflect the explanation faithfulness. Just curious whether there are any other additional explanation-specific metrics?

Q2: In Figure 2., in the right side, are the stability graph and the transition graphs share the same encoder? And also the same one as the input temporal graph encoder?

Q3: I am still unclear about how unseen interaction or first interaction can be thoroughly decomposed into these two patterns? Also, how should we understand Figure 10? Is it more like separately visualized stability and transition patterns? How should we understand the difference between them?

---

> ### Author Rebuttal · Authors · 2026-03-31
>
> We thank the reviewer for positively assessing our method’s novelty, theoretical grounding, and empirical results. Below, we address the raised concerns:
>
> **Q1: Explanation Sparsity and Metrics**
>
> *Explanation sparsity* is the fraction of edges retained in the explanation subgraph relative to the full input. At sparsity level $s \in [0, 0.3]$, we keep the top-$s$ fraction of edges ranked by importance scores. Our ACC-AUC evaluation follows the protocol in TempME[1]: we compute the agreement between the subgraph’s prediction and the full model’s prediction across sparsity levels in this range, and integrate it into a single score (formal definition in Appendix E).
>
> ACC-AUC primarily measures self-consistency (whether the subgraph reproduces the full model's prediction), not ground-truth faithfulness. To directly evaluate explanation fidelity, we adopt AUFSC (Area Under the Fidelity-Sparsity Curve) from the CoDy protocol[2], which integrates fidelity across sparsity levels into a single $[0,1]$ score:
>
> - **AUFSC$^+$** evaluates **necessity**: removing the identified important edges should change the prediction. Higher AUFSC$^+$ = the explanation correctly identifies edges the model depends on.
> - **AUFSC$^-$** evaluates **sufficiency**: keeping *only* the identified important edges should preserve the prediction. Higher AUFSC$^-$ = the explanation captures enough evidence to support the prediction.
>
> We further report AUFSC separately on correctly and incorrectly predicted instances. Results:
>
> |Method|AUFSC$^+$(Correct) Wikipedia/UCI|AUFSC$^+$(Incorrect) Wikipedia/UCI|AUFSC$^-$(Correct) Wikipedia/UCI|AUFSC$^-$(Incorrect) Wikipedia/UCI|
> |-|-|-|-|-|
> |PGExplainer|0.03/0.02|0.11/0.08|0.67/0.39|0.54/0.61|
> |TGNNExplainer|0.01/0.05|0.10/0.14|0.61/0.45|0.43/0.53|
> |TempME|0.04/0.06|0.15/0.22|0.72/0.56|0.50/0.84|
> |TGIB|0.04/0.04|0.18/0.20|0.69/0.49|0.41/0.78|
> |CoDy|0.15/0.16|0.21/0.39|0.82/0.65|0.54/0.87|
> |Ours|0.15/0.22|0.24/0.39|0.91/0.86|0.62/0.91|
>
> Our method achieves the best or tied-best AUFSC$^+$ and significantly outperforms on AUFSC$^-$ (e.g., UCI Correct: 0.86 vs. CoDy 0.65), confirming that our explanations identify edges that are both necessary and sufficient for the model's predictions. Due to space, we report two representative datasets here and will include full results in the revised version.
>
> **Q2: Shared Encoder Architecture**
>
> Yes, $G_S$, $G_T$, and the input $G^t_{u,v}$ share the same $\text{Encoder}\theta$ (Eq. 11, 18). This serves two purposes: (1) *parameter efficiency*—avoiding tripling model parameters; (2) *representational alignment*—encoding $G_S$ and $G_T$ in the same latent space is essential for the adversarial $L_{Dis}$ objective (Eq. 19), which enforces conditional independence between $h_S$ and $h_T$ given $Y$. The disentanglement arises from frequency-based soft masking (producing distinct input subgraphs) combined with $L_{Dis}$ (pushing representations apart), not from separate encoders.
>
> **Q3: Decomposition of Unseen Interactions and Figure 10**
>
> The stability/transition decomposition is applied to the historical neighborhood subgraph $G_E$ (one-hop ego-history around $u$ and $v$), not to the target edge itself. Thus, even when the target pair $(u,v)$ is unseen before $t$, its historical neighborhood still contains edges with different recurrence support; the soft assignment in Eq.(16–17) is computed per neighborhood edge to form $G_S$ and $G_T$.
>
> Figure 10 presents two case studies: (a) a stability interaction (previously seen pair) and (b) a transition interaction (previously unseen pair). For each, three methods select 5 explanatory edges. ST-TGExplainer selects different evidence structures for the two types: in (a), it focuses on temporally concentrated, recurrent evidence around the target pair; in (b), it selects a broader set of neighbor connections capturing the emerging context. Note that these subgraphs are the **final explanation output**, not the internal $G_S/G_T$ decomposition.
>
> [1] TempME: Towards the Explainability of Temporal Graph Neural Networks via Motif Discovery. NeurIPS 2023.
>
> [2] CoDy: Counterfactual Explainers for Dynamic Graphs. ICML 2025.

---

> > ### Author Rebuttal · Reviewer_27S1 · 2026-04-01
> >
> > The rebuttal has adequately addressed my questions. My assessment remains positive, and I am satisfied with the authors' responses. I maintain my score.

---

> > > ### Author Response · Authors · 2026-04-03
> > >
> > > Thank you for the follow-up and for maintaining your positive assessment. We appreciate your careful review and constructive feedback. We will incorporate the clarifications and revisions outlined in our rebuttal into the final version.

---

### Official Review · Reviewer_FvXC · 2026-03-13

**Soundness:** 3
**Presentation:** 3
**Significance:** 2
**Originality:** 3
**Overall Recommendation:** 4
**Confidence:** 5

**Summary:**

This paper proposes ST-TGExplainer, a self-explainable temporal GNN for link prediction that separates two types of evidence in temporal graphs: stability patterns from repeated historical interactions and transition patterns from less frequent or newly emerging interactions. The method combines an informative interaction selector, a stability-transition graph disentangler, and a pattern feature ensembler, and is trained with a disentangled information bottleneck objective to extract compact explanatory subgraphs while reducing redundancy between the two pattern types. Experiments on six temporal graph datasets show strong link prediction performance and improved explanation ACC-AUC relative to several TGNN and explainer baselines.

**Compliance With Llm Reviewing Policy:**

Affirmed.

**Final Justification:**

As stated in the rebuttal acknowledgement, my major concerns are addressed, so I raised my score from 3 to 4.

**Key Questions For Authors:**

1. How exactly are transition patterns defined in the method? In my understanding, they should be the neighbors (1-hop or 2-hop?) of the target that appear for the first time in the entire graph. The ‘interaction frequency between node pairs’ in 4.3 sounds odd to me.
2. How do you control for backbone effects in the explanation comparison? And can you apply the baseline explanation methods to your new model? Some methods are model-agnostic, so this should be possible. Maybe you can also evaluate the explanations separately for correct and incorrect predictions.
3. Can you report results for other evaluation metrics such as sparsity, fidelity+/fidelity-, or an evaluation restricted to correctly predicted instances? If the method remains superior, that would strengthen the empirical case.
4. Can you also report the runtime of the model?

**Limitations:**

Yes, the limitations are discussed in the conclusion.

**Strengths And Weaknesses:**

**Strengths**
- The distinction between stability and transition evidence is a reasonable and potentially useful perspective for temporal explanation. This framing is intuitive and helps motivate why repeated and non-repeated interactions may play different roles.
- The method is intrinsically explainable rather than purely post-hoc, which is appealing.
- The paper includes a fairly broad empirical study, with six datasets, multiple link prediction baselines, and multiple explanation baselines.
- The method appears competitive in predictive performance, especially on datasets with lower repeat ratios.
- The paper is reasonably well structured and written.

**Weaknesses**
- The paper first frames transition patterns as “newly emerging first-time interactions,” but then in section 4.3, it says “leveraging the empirical interaction frequency between node pairs”. Low frequency is not equivalent to newly emerging interactions: an interaction can be rare but previously observed, while the first occurrence that later becomes frequent is still a transition event conceptually.
- The explainability study is not a fair comparison. The baselines are based on GraphMixer, while the proposed method is evaluated on its own self-explainable model. The results may only reflect model difference rather than explanation quality, as shown in table 2, GraphMixer is one of the weak baselines in link prediction. It would be better to evaluate them separately for correct and incorrect predictions, following the CoDy setup. Moreover, it would be interesting to test the explainability baselines on the newly proposed model.
- The explanation metric is too narrow, it is not enough to support faithfulness alone. It would be better if further metrics such as sparsity, fidelity+, and fidelity- are also reported.
- Figure 1 is not clear enough. It needs precise explanation on how the categories are defined and how many test interactions fall into each category.
- Figure 6 is weak. PCA does not provide enough information for the disentanglement.
- I think CoDy is an important baseline for the explanation study. It is discussed in the related work, but not included in the experiments.
- The paper does not appear to provide code or an anonymous repository.

---

> ### Author Rebuttal · Authors · 2026-03-31
>
> We thank the reviewer for their time and comments. We address the raised concerns below:
>
> **W1: Transition Definition vs. Frequency Proxy**
>
> In our paper, transition is conceptually aligned with first-time/newly emerging interactions. Low frequency is not strictly equivalent to “first-time emergence.” Sec.4.3 adopts a frequency-based soft assignment as a differentiable surrogate to approximate emergence strength, because hard first-time indicators are brittle and non-differentiable under neighbor sampling. This approximation can assign transition weight to rare-but-seen interactions. We will revise the introduction and Sec.4.3 to explicitly state this surrogate relationship (approximation, not redefinition) and clarify the resulting boundary cases.
>
>
> **W2/Q2: Shared-Backbone Baselines and CoDy-Style Evaluation**
>
> We would be happy to make a fairer comparison of the models’ explanations. We therefore (1) run model-agnostic post-hoc explainers on our trained predictor/backbone (MLP-Mixer–based encoder in Sec.4.2) under the same ACC-AUC protocol, with results shown below:
> |Models|Wikipedia|UCI|
> |-|-|-|
> |GNNExplainer|87.21±0.73|63.02±0.77|
> |PGExplainer|86.19±1.34|62.76±1.36|
> |T-GNNExplainer|89.69±0.96|82.47±0.87|
> |TempME|89.15±0.40|90.87±0.52|
> |TGIB|88.09±0.68|87.06±1.04|
> |CoDy|91.39±0.54|90.28±0.75|
> |Ours|92.47±0.26|96.04±0.39|
>
> and (2) following the CoDy setup, report AUFSC$^+$/AUFSC$^-$ with correct/incorrect predictions (see **Reviewer 27S1 Q1**). Under both same-backbone ACC-AUC and AUFSC (correct/incorrect) evaluations, our advantage persists, supporting that gains reflect explanation quality rather than backbone choice.
>
> **W3/Q3: Additional Evaluation Metrics**
>
> Following CoDy, we integrate fidelity across sparsity: AUFSC$^+$ removes selected edges (necessity/fidelity+) and AUFSC$^-$ keeps only selected edges (sufficiency/fidelity−). We sweep sparsity in the same range; results on Wikipedia/UCI (**see Reviewer 27S1 Q1**) show ours is best/tied on AUFSC$^+$ and best on AUFSC$^-$.
>
> **W4: Fig.1 Clarity**
>
> Following the TGB-Seq protocol, an interaction $(u,v,t)$ is classified as seen if the node pair $(u,v)$ appeared at least once before time $t$, and unseen otherwise. We will add this definition and the test-set breakdown to the caption:
> |Dataset|\#Test|\#Seen|\#Unseen|Unseen%|
> |:-|:-|:-|:-|:-|
> |Wikipedia|23,621|21,077|2,544|10.77%|
> |Reddit|100,867|93,873|6,994|6.93%|
> |UCI|8,976|6,399|2,577|28.71%|
> |Enron|18,785|16,732|2,053|10.93%|
> |USLegis|4,950|3,268|1,682|33.98%|
> |Can.Parl.|10,113|4,615|5,498|54.37%|
>
> Fig.1 shows a consistent MRR drop on unseen pairs, motivating explicitly modeling and explaining evidence for emerging interactions (with unseen pairs being a representative case), rather than relying predominantly on recurrent evidence.
>
> **W5: Fig.6 PCA and Disentanglement Evidence**
>
> To provide stronger evidence beyond PCA, we add an ablation that splits $G_E$ into ($G_S$,$G_T$). $G_E$-only removes the split and predicts from $G_E$; $G_S$-only/$G_T$-only use one branch. Results:
> |Variant|Wikipedia AP/ACC-AUC|UCI AP/ACC-AUC|
> |-|-|-|
> |Fullmodel|99.21/92.47|97.28/96.04|
> |$G_E$-only|98.14/88.58|96.06/88.06|
> |$G_S$-only|98.01/87.93|91.76/84.67|
> |$G_T$-only|96.36/77.61|93.62/85.18|
>
> The full model consistently outperforms $G_E$-only by 3.89–7.98 ACC-AUC points, indicating that the split provides complementary evidence beyond using a single mixed explanation subgraph. Moreover, $G_S$-only is stronger on Wikipedia (high recurrence), while $G_T$-only is relatively stronger on UCI (lower recurrence), consistent with the intended roles of stability vs. transition evidence.
>
>
> **W6: CoDy as Baseline**
>
> CoDy is included in the shared-backbone ACC-AUC table in **W2**. AUFSC$^+$/AUFSC$^-$ comparisons (**Reviewer 27S1 Q1**) also favor ours, especially on AUFSC$^-$.
>
> **W7: Code Availability**
>
> Anonymous repository: https://anonymous.4open.science/r/ST-TGExplainer
>
> **Q1: Definition of Transition Patterns**
>
> We use a 1-hop neighbor. For a target $(u,v,t)$, we collect historical edges incident to $u$ or $v$ before $t$ and keep the $L$ most recent ones, forming $G_E$ (Sec.4.2). Conceptually, transition patterns correspond to newly emerging / first-time interactions. In implementation (Sec.4.3), we use a frequency-based soft assignment as a differentiable surrogate: compute pair co-occurrence counts $h_F$, map to $p_f$, and split $G_E$ into $G_S$ and $G_T$ via $p_f$. As in **W1**, this is an approximation for optimization when the hard indicator is *non-differentiable*; we will revise wording accordingly.
>
> **Q4: Runtime**
>
> Explanation generation time on Wikipedia/UCI is reported below. For link prediction running time, Fig.3 reports training time per epoch and parameter size, showing competitive efficiency with a compact footprint.
> |Method|Wikipedia|UCI|
> |-|-|-|
> |T-GNNExplainer|68.6|68.5|
> |TempME|75.4|88.4|
> |TGIB|120.1|147.5|
> |CoDy|105.0|115.8|
> |Ours|53.1|66.7|

---

> > ### Author Rebuttal · Reviewer_FvXC · 2026-04-01
> >
> > Thank you for the detailed rebuttal and additional experimental results.
> >
> > Overall, the rebuttal meaningfully strengthens the empirical evaluation and clarifies several methodological points. In particular, the additional shared-backbone comparisons, the inclusion of CoDy, and the AUFSC-based evaluation help address my concerns regarding explanation fairness and evaluation metrics.
> >
> > Some conceptual questions remain regarding the alignment between the conceptual definition of transition patterns (as newly emerging interactions) and their frequency-based approximation in the model.
> >
> > Nevertheless, the rebuttal improves my confidence in the empirical results, and I therefore update my assessment slightly upward.

---

> > > ### Author Response · Authors · 2026-04-03
> > >
> > > Thank you for your thoughtful follow-up and for updating your assessment. We appreciate your positive remarks on the additional shared-backbone comparisons, the inclusion of CoDy, and the AUFSC-based evaluation. Regarding the remaining conceptual point, the frequency-based term in Sec. 4.3 is intended as a differentiable surrogate for optimization, rather than a redefinition of transition as strictly "first-time" events; as noted, it can assign some transition weight to rare-but-seen interactions. We will make this surrogate relationship and the relevant boundary cases explicit in the revised paper and align the wording accordingly. Thank you again for your careful review and constructive discussion.

---

### Decision · Program_Chairs · 2026-04-30

**Decision:**

Accept (regular)

**Comment:**

After the rebuttal, all reviewers agree that this is a technically sound, well-written and novel contribution that is relevant to some part of the ICML community, and thus should be accepted. The initial reviews raised several concerns, but they were all properly addressed in the rebuttal. The authors should incorporate in the revision  all elements that were key to address these concerns in the rebuttals.